# Endothelial cells regulate alveolar morphogenesis by constructing basement membranes acting as a scaffold for myofibroblasts

Alveologenesis is a spatially coordinated morphogenetic event, during which alveolar myofibroblasts surround the terminal sacs constructed by epithelial cells and endothelial cells (ECs), then contract to form secondary septa to generate alveoli in the lungs. Recent studies have demonstrated the important role of alveolar ECs in this morphogenetic event. However, the mechanisms underlying EC-mediated alveologenesis remain unknown. Herein, we show that ECs regulate alveologenesis by constructing basement membranes (BMs) acting as a scaffold for myofibroblasts to induce septa formation through activating mechanical signaling. Rap1, a small GTPase of the Ras superfamily, is known to stimulate integrin-mediated cell adhesions. EC-specific *Rap1*-deficient (*Rap1^{iECKO}*) mice exhibit impaired septa formation and hypo-alveolarization due to the decreased mechanical signaling in myofibroblasts. In *Rap1^{iECKO}* mice, ECs fail to stimulate integrin β1 to recruit Collagen type IV (Col-4) into BMs required for myofibroblast-mediated septa formation. Consistently, EC-specific integrin β1-deficient mice show hypo-alveolarization, defective mechanical signaling in myofibroblasts, and disorganized BMs. These data demonstrate that alveolar ECs promote integrin β1-mediated Col-4 recruitment in a Rap1-dependent manner, thereby constructing BMs acting as a scaffold for myofibroblasts to induce mechanical signal-mediated alveologenesis. Thus, this study unveils a mechanism of organ morphogenesis mediated by ECs through intrinsic functions.

Alveologenesis is the final step of lung development aimed at generating alveoli, creating a large surface area for gas exchange between the lungs and blood. In mice, alveologenesis starts after birth and continues for several weeks, ultimately giving rise to more than 2 million alveoli. Alveolar formation is a spatially coordinated morphogenetic event regulated by multiple cell types such as alveolar epithelial type I (AT1) and type II (AT2) cells, myofibroblasts, endothelial cells (ECs), pericytes, and autonomic neurons[1–4]. During alveologenesis, the walls of terminal sacs at the ends of the respiratory bronchioles protrude into the air space to form secondary septa, which subdivide at the terminal sac to generate alveoli. Two-dimensional (2D) analysis of terminal sacs revealed that myofibroblasts exist at the tip of a growing secondary septum. Furthermore, a reduction of myofibroblasts or their specific ablation in mice reportedly prevented alveolar formation[5–8]. Thus, myofibroblasts are regarded as a key player in secondary septa formation. Importantly, Sun's group, by performing 3D imaging analysis of alveolar morphogenesis, showed that thinly stretched and long myofibroblasts surround the alveolar entry rings

✉ e-mail: t-haruko@nms.ac.jp; s-fukuhara@nms.ac.jp

and contract against air pressure on the alveolar wall to induce the formation of secondary septa[9]. Furthermore, recent studies have shown that myofibroblast contraction required for septa formation depends on insulin-like growth factor 1 receptor-stimulated mechanical signaling that involves myosin light chain kinase (MLCK)-mediated activation of myosin and nuclear localization of Yes-associated protein (YAP), an effector of the Hippo pathway[10,11].

ECs not only form blood vessel networks for delivering blood but also produce paracrine cues referred to as angiocrine factors, which actively regulate the development, maintenance and regeneration of various organs, the self-renewal and differentiation of stem cells, and even underlie the pathogenesis of many diseases such as tissue fibrosis and tumor progression[12–14]. Indeed, pulmonary EC-derived angiocrine factors were shown to regulate lung development and regeneration. EC-derived hepatocyte growth factor stimulates epithelial cells to induce primary septa formation during distal lung morphogenesis[15]. Previous reports also showed that upon pneumonectomy, pulmonary ECs promote the proliferation of alveolar epithelial cells and bronchioalveolar stem cells (BASCs) by releasing matrix metalloproteinase 14 to increase the bioavailability of EGF ligands[16,17]. Furthermore, during lung regeneration, BASCs were shown to be specified into alveolar epithelial cells by thrombospondin-1 produced by pulmonary ECs[18]. In addition, several studies have clearly shown the essential role of pulmonary ECs in alveologenesis. Inhibition of vascular endothelial growth factor (VEGF) signaling after birth decreases alveolarization possibly through prevention of angiogenesis[19,20]. Furthermore, mice lacking platelet endothelial cell adhesion molecule-1 (PECAM1) reportedly exhibit impaired alveologenesis, pointing to an active role of ECs in alveolarization[21]. Importantly, a recent study demonstrated that loss of aerocytes (aCap), alveolar capillary ECs specialized for gas exchange, resulted in hypo-alveolarization despite normal differentiation of myofibroblasts[22]. These results suggest that pulmonary ECs regulate alveolar morphogenesis independently of inducing myofibroblast differentiation. However, the mechanism underlying EC-mediated alveologenesis is still not fully understood.

Herein, we analyzed EC-specific Rap1a/Rap1b double knockout mice and uncovered a molecular mechanism governing the regulation of alveolar morphogenesis by pulmonary ECs. Rap1, a small GTPase belonging to the Ras superfamily, is known to potentiate both integrin-mediated cell adhesions into the extracellular matrix and cadherin-mediated cell-cell junctions[23–27]. Mammalian genomes encode two highly homologous and evolutionarily conserved Rap1 genes, Rap1a and Rap1b. We found that postnatal deletion of endothelial Rap1a and Rap1b resulted in impairment of mechanical signaling in alveolar myofibroblasts characterized by compromised alveologenesis. The link between endothelial Rap1 and mechanical activation of alveolar myofibroblasts is mediated via endothelial Rap1 stimulation of integrin β1, which in turn induces the formation of basement membranes (BMs) that act as a scaffold for myofibroblasts and promote alveologenesis through mechanical signaling. Thus, this study unveils a mechanism of organ morphogenesis mediated by ECs, which affect myofibroblasts by assembling a BM rather than producing angiocrine factors.

## Results

### Rap1 is required for alveologenesis

To investigate the role of Rap1 in ECs during early postnatal stages, we generated mice lacking both Rap1a and Rap1b in ECs (Rap1[iECKO]) by crossing double-floxed Rap1a/b mice (Rap1[fl/fl]) with mice expressing CreERT2 recombinase under the control of the Cdh5 promoter and by delivering tamoxifen via the lactating mothers for three consecutive days from postnatal day 0 (P0) (Supplementary Fig. 1a). Rap1[fl/fl] mice without the CreERT2 gene served as a control. To ensure the specific ablation of Rap1a/b paralogs in the EC lineage, we isolated CD31 (also

known as PECAM1)-positive [CD31(+)] ECs and platelet-derived growth factor receptor α (PDGFRα)-positive [PDGFRα(+)] cell populations including myofibroblasts from the lungs of P9 pups by fluorescence-activated cell sorting (FACS). CD31(+) and PDGFRα(+) cells were found to express the EC marker, Pecam1, and the myofibroblast marker, Eln, respectively, confirming sorting of the desired cell populations (Supplementary Fig. 1b). Importantly, expression levels of Rap1a and Rap1b mRNAs in CD31(+) cells derived from Rap1[iECKO] mice were significantly downregulated compared to those from Rap1[fl/fl] mice (Supplementary Fig. 1c). However, their expression was not altered in PDGFRα(+) cells regardless of genotypes, confirming EC-specific and efficient deletion of Rap1a and Rap1b genes in Rap1[iECKO] mice.

We analyzed Rap1[iECKO] mice and found that they exhibited postnatal lethality. Rap1[iECKO] mice started to die at P15 and more than 90% of the mutant mice were lost by P30 (Fig. 1a). When identifying the causes of death, we noticed that the lungs of Rap1[iECKO] mice were significantly smaller than those of controls at P21 (Fig. 1b, c). This difference in lung size was already apparent at P14, but not at P7, even though the body weights of control and Rap1[iECKO] mice were similar (Supplementary Fig. 1d, e). Furthermore, Rap1[iECKO] mice exhibited an increased mean linear intercept (MLI) of the lungs compared to control siblings at P14, but not at P7 (Fig. 1d, e and Supplementary Fig. 1f), indicating that postnatal alveologenesis is compromised in the absence of endothelial Rap1. On the other hand, tissue morphologies of the liver, stomach, and intestine appeared to be normal at P14 in Rap1[iECKO] mice, suggesting that the lung abnormality is not secondary to the systemic defects (Supplementary Fig. 1g).

Next, we investigated whether defective alveologenesis in Rap1[iECKO] mice is attributable to abnormal development of the lung vasculature by performing 3D immunofluorescence analysis of lungs using anti-CD31 and anti-intercellular adhesion molecule 2 (ICAM2) antibodies. However, the alveolar vascular structure in Rap1[iECKO] mice at P4 and P9 was almost identical to that in control siblings (Fig. 1f and Supplementary Fig. 1h). The alveolar capillary network is composed of two types of ECs, the aCap and the general capillary (gCap) ECs[22]. Staining of P14 lungs with antibodies for their specific marker carbonic anhydrase 4 (Car4) for aCap and plasmalemma vesicle-associated protein (PLVAP) for gCap showed that aCap and gCap ECs were in normal amounts and locations in the alveolar capillaries of Rap1[iECKO] mice (Supplementary Fig. 1i). Furthermore, staining of P9 lungs with antibodies for the Ets-related gene (ERG) (a marker of endothelial nuclei), for phosphorylated histone H3 (PHH3) (a proliferation marker), and for Cleaved-caspase3 (an apoptosis marker) revealed similar numbers of alveolar capillary ECs as well as the frequency of proliferation and apoptosis in P9 lungs of control and Rap1[iECKO] mice (Supplementary Fig. 1j–m). These results indicate that alveolar capillary networks develop normally in Rap1[iECKO] mice. Rap1 is known to potentiate endothelial barrier function by forming vascular endothelial (VE)-cadherin-mediated EC-EC junctions[24,28–30]. However, Rap1[iECKO] mice did not exhibit vascular hemorrhage (Supplementary Fig. 1n), suggesting the barrier function of alveolar capillaries to be preserved in those mice. Collectively, our findings suggest that alveolar capillary ECs actively regulate alveolar morphogenesis in a Rap1-dependent manner.

### Rap1 regulates mechanical signaling required for alveolar myofibroblast contraction

Postnatal alveolar morphogenesis depends largely on the contractile force produced by alveolar myofibroblasts[9]. Hence, to investigate whether alveolar myofibroblast differentiation is affected in Rap1[iECKO] mice, we employed 3D immunostaining with a myofibroblast marker antibody [α-smooth muscle actin (α-SMA)] and found that myofibroblasts were present in terminal sacs at P4 and localized to the alveolar entry ring at P9 in both control and Rap1[iECKO] mice (Fig. 2a and

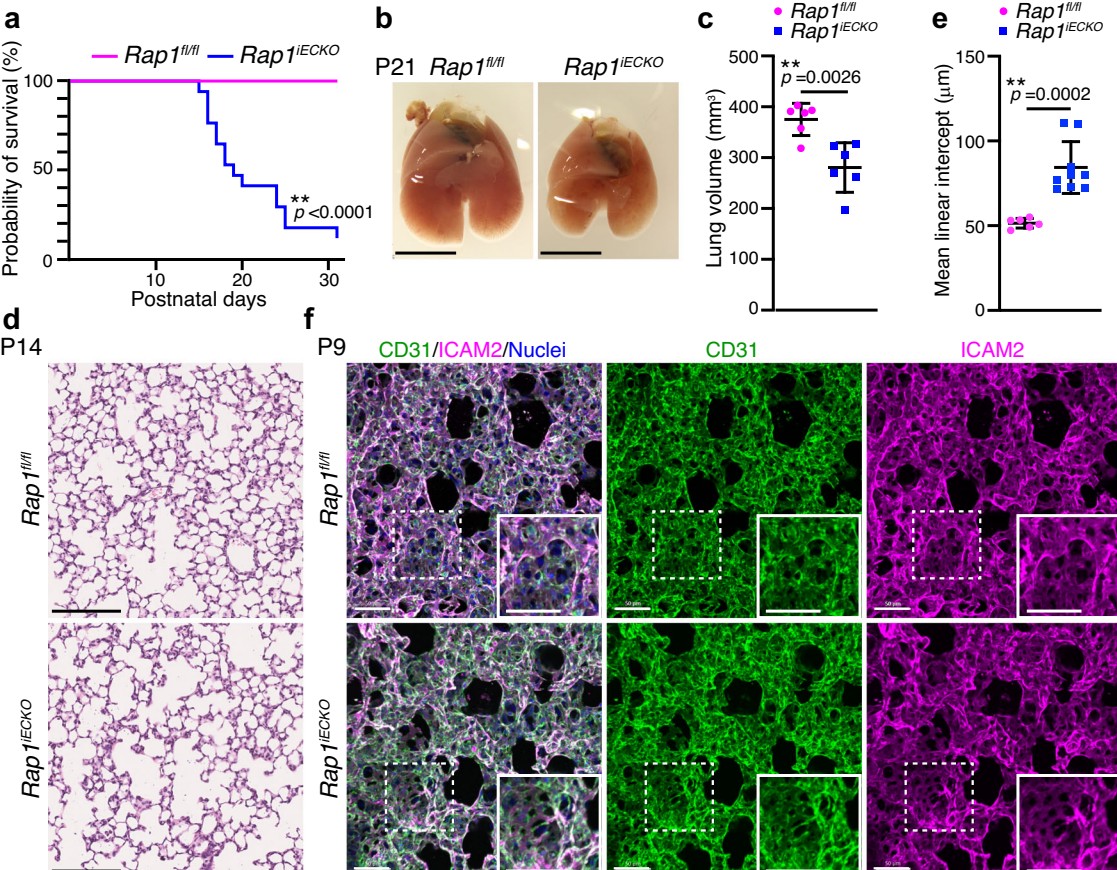

**Fig. 1 | Endothelial Rap1 is required for alveologenesis. a** Kaplan–Meier survival curves for *Rap1^{fl/fl}* (pink) and *Rap1^{iECKO}* (blue) mice (*n* = 22 mice/each). **b** Representative images of the lungs from *Rap1^{fl/fl}* and *Rap1^{iECKO}* mice at P21. **c** Lung volumes of *Rap1^{fl/fl}* and *Rap1^{iECKO}* mice at P21. Each dot represents an individual mouse. Data are means ± s.d. (*n* = 6 mice/each). **d** Representative images of HE-stained lung tissues from *Rap1^{fl/fl}* and *Rap1^{iECKO}* mice at P14. **e** Mean linear intercepts (MLI) in the lungs of *Rap1^{fl/fl}* and *Rap1^{iECKO}* mice at P14, as in (**d**). Each dot represents

an individual mouse. Data are means ± s.d. (*Rap1^{f/f}* and *Rap1^{iECKO}*, *n* = 6 and 9 mice). **f** Confocal z-projection images for CD31 (green), ICAM2 (magenta), and Nuclei (DAPI, blue) in alveoli isolated from *Rap1^{fl/fl}* and *Rap1^{iECKO}* mice at P9. Boxed areas are enlarged in the insets. **P < 0.01, by the log-rank test (**a**) and two-tailed Student's *t*-test (**c**, **e**). Scale bars; 5 mm (**b**), 250 μm (**d**), 50 μm (**f**). Source data are provided as a Source data file.

Supplementary Fig. 2a). Consistently, autonomic nerves, which have recently been shown to regulate myofibroblast proliferation and migration[4], developed normally in *Rap1^{iECKO}* mice (Supplementary Fig. 2b). Flow cytometric analysis also revealed normal numbers of alveolar myofibroblasts [α-SMA(+)/PDGFRα(+) cells], myofibroblast progenitors [PDGFRα(+) cells], and smooth muscle cells [α-SMA(+) cells] in the lungs of *Rap1^{iECKO}* mice (Fig. 2b). These findings indicate that myofibroblast differentiation occur normally in *Rap1^{iECKO}* mice. We also found that the appearance of AT1 cells labeled with their marker antibody [Advanced Glycosylation End-Product Specific Receptor (RAGE)], did not differ between control and *Rap1^{iECKO}* mice at either P4 or P9 (Supplementary Fig. 2c), revealing that AT1 cells are not involved in the impairment of alveologenesis in *Rap1^{iECKO}* mice.

However, we observed the α-SMA filaments in myofibroblasts to be markedly thinner in *Rap1^{iECKO}* mice than in controls (Fig. 2c, d and Supplementary Movie 1, 2), pointing to a functional impairment of myofibroblasts in *Rap1^{iECKO}* mice. Alveolar myofibroblasts generate contractile force through activation of mechanical signaling involving MLCK-dependent phosphorylation of myosin light chain (MLC) and nuclear localization of YAP[10,11]. Therefore, to investigate whether mechanical signaling in alveolar myofibroblasts is affected in *Rap1^{iECKO}* mice, we employed 3D immunostaining with antibodies against phosphorylated-myosin light chain (pMLC) and YAP. In control mice, myofibroblasts localized to the alveolar entry ring showed a strong pMLC signal (Fig. 2e, f). However, the pMLC signal in alveolar

myofibroblasts of *Rap1^{iECKO}* siblings was dramatically decreased. Furthermore, the number of alveolar myofibroblasts exhibiting the nuclear YAP signal in *Rap1^{iECKO}* mice was also significantly lower than that in control mice (Fig. 2g, h). Cells other than myofibroblasts in the lungs of *Rap1^{iECKO}* mice also tended to have a weak nuclear YAP signal. Consistently, YAP protein levels in the lungs of *Rap1^{iECKO}* mice were lower than those in control mice, suggesting reduced nuclear YAP localization in the *Rap1^{iECKO}* mice, since YAP retained in the cytoplasm undergoes proteasomal degradation[31] (Fig. 2i, j). These findings indicate that mechanical signaling in alveolar myofibroblasts is impaired in *Rap1^{iECKO}* mice.

Next, we investigated whether mechanical signaling in myofibroblasts of *Rap1^{iECKO}* mice is compromised in a cell-autonomous or a non-cell-autonomous manner. To this end, PDGFRα(+) cells including myofibroblasts and their progenitors were sorted by FACS from P8-9 lungs of control and *Rap1^{iECKO}* mice, and subjected to collagen gel contraction assay. The contractile ability of PDGFRα(+) cells derived from *Rap1^{iECKO}* mice was similar to that of control mouse-derived cells (Fig. 2k, l). Furthermore, PDGFRα(+) cells derived from *Rap1^{iECKO}* mice and those from control mice exhibited a similar degree of nuclear YAP localization when plated on a glass-base dish (Supplementary Fig. 2d, e), indicating the mechanotransduction machinery to be preserved even in the *Rap1^{iECKO}* mouse-derived cells. These results suggest that mechanical signaling in myofibroblasts is non-cell-autonomously impaired in *Rap1^{iECKO}* mice.

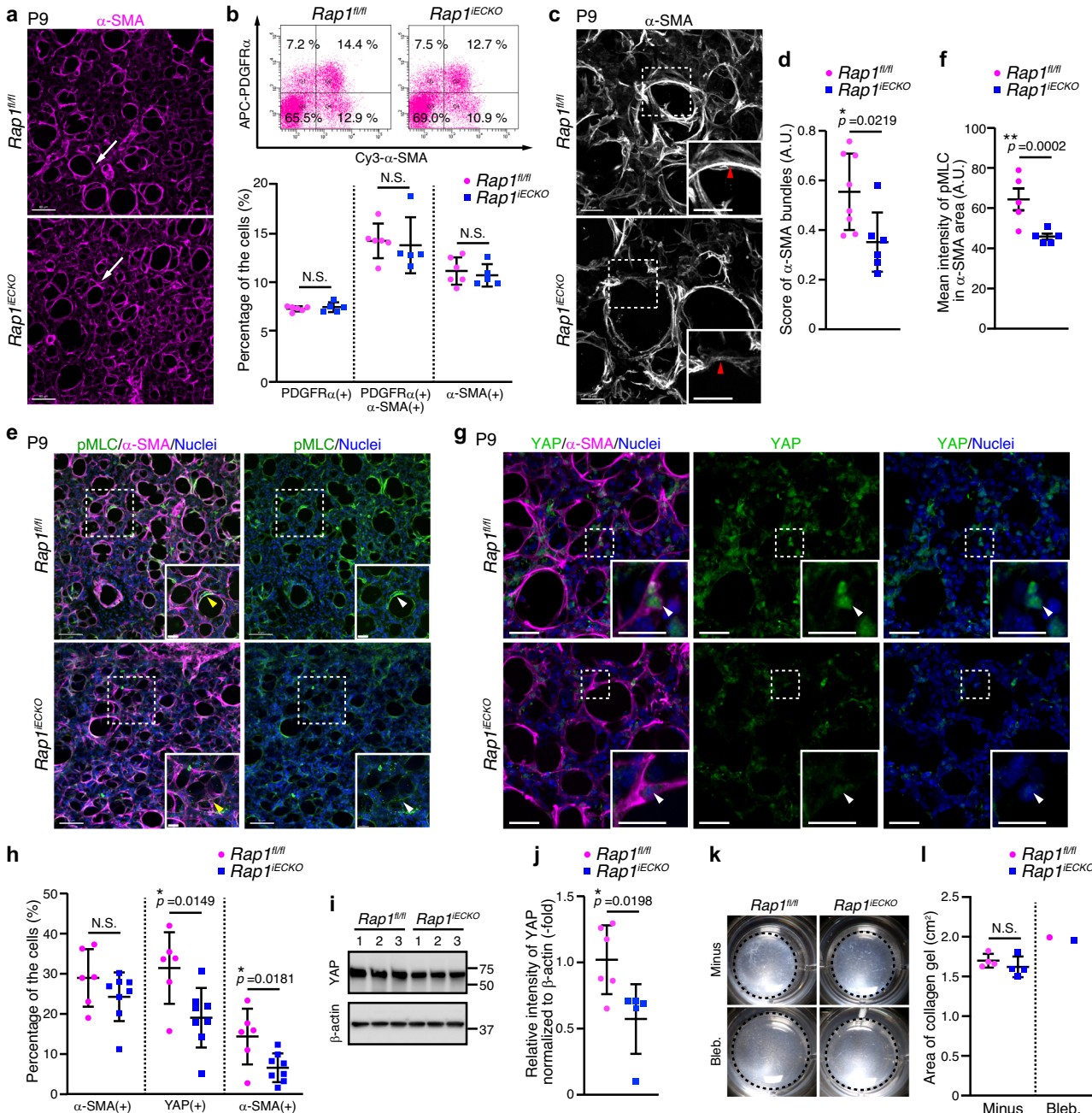

**Fig. 2 | Endothelial Rap1 regulates mechanical signaling required for alveolar myofibroblast contraction. a** Confocal z-projection images for α-SMA in alveoli from P9 *Rap1^{fl/fl}* and *Rap1^{iECKO}* mice. Arrowheads, myofibroblasts surrounding the alveolar entry ring. **b** Upper, flow cytometry plots of myofibroblasts [α-SMA(+)/PDGFRα(+) cells], their progenitors [PDGFRα(+) cells] and smooth muscle cells [α-SMA(+) cells] in the lungs of P9 *Rap1^{fl/fl}* and *Rap1^{iECKO}* mice. Plots show Cy3-anti-α-SMA versus APC-anti-PDGFRα fluorescence. Lower, percentages of cell populations as indicated at the bottom (*Rap1^{f/f}* and *Rap1^{iECKO}*, *n* = 6 and 5 mice). **c** Confocal z-projection images for α-SMA in alveoli from P9 *Rap1^{fl/fl}* and *Rap1^{iECKO}* mice. Arrowheads, α-SMA filaments. **d** Score of α-SMA bundles in alveoli (arbitrary units, A.U.), as in (**c**) (*Rap1^{f/f}* and *Rap1^{iECKO}*, *n* = 8 and 6 mice). **e** Confocal z-projection images for pMLC(S19) (green), α-SMA (magenta), and Nuclei (DAPI, blue) in alveoli from P9 *Rap1^{fl/fl}* and *Rap1^{iECKO}* mice. Arrowheads, pMLC signal in myofibroblasts. **f** pMLC fluorescence intensity in α-SMA(+) areas (arbitrary units, A.U.), as in (**e**) (*n* = 5 mice/each). **g** Confocal z-projection images for YAP (green), α-SMA

(magenta), and Nuclei (DAPI, blue) in alveoli from P9 *Rap1^{fl/fl}* and *Rap1^{iECKO}* mice. Arrowheads, the nuclei of α-SMA(+) myofibroblasts. **h** Percentages of α-SMA(+), nuclear YAP(+), α-SMA(+)/nuclear YAP(+) cell populations, as in (**g**) (*Rap1^{f/f}* and *Rap1^{iECKO}*, *n* = 6 and 8 mice). **i** Western blot analysis of YAP and β-actin in lungs of *Rap1^{fl/fl}* and *Rap1^{iECKO}* mice at P9 (3 mice/each). **j**, Expression levels of YAP relative to those of β-actin, as in **i** (*Rap1^{f/f}* and *Rap1^{iECKO}*, *n* = 6 and 5 mice). **k** Representative images of collagen gels containing PDGFRα(+) cells from the lungs of P9 *Rap1^{fl/fl}* and *Rap1^{iECKO}* mice and cultured without (Minus) or with 10 μM blebbistatin (Bleb.). **l**, Quantification of collagen gel areas (*Rap1^{f/f}* and *Rap1^{iECKO}* without/with Bleb., *n* = 6/1 and 6/1 mice). Each dot represents an individual mouse, and data are presented as means ± s.d. (**b**, **d**, **f**, **h**, **j**, **l**). N.S., no significance; *P* < 0.05 by two-tailed Student's *t*-test (**d**, **f**, **h**, **j**, **l**). Boxed areas are enlarged in the insets (**c**, **e**, **g**). Scale bars; 80 μm (**a**, **e**), 20 μm (**c**, **g** and enlarged in **c** and **e**), 10 μm (enlarged in **g**). Source data are provided as a Source data file.

## ECs induce Col-4 accumulation to generate BMs acting as a scaffold for myofibroblasts in a Rap1-dependent manner

The next question to address was how ECs regulate mechanical signaling in alveolar myofibroblasts in a Rap1-dependent manner. To tackle this issue, we first analyzed spatial localizations of ECs and myofibroblasts in alveoli (Supplementary Fig. 3a). Capillary ECs surrounded and created intimate contacts with AT1 cells to form alveoli in P9 lungs. Furthermore, myofibroblasts covered the outside of the alveolar space by making tight contacts with ECs. It can reasonably be assumed that BMs mediate an indirect contact between ECs and myofibroblasts. In addition, BMs are known to act as scaffolds for cells to regulate development and tissue morphogenesis[32–34]. Importantly, the stiffness of the extracellular matrix forming BMs has also been shown to stimulate mechanical signaling involved in various cellular functions[35,36]. Therefore, we hypothesized that ECs might regulate alveologenesis by constructing BMs acting as a scaffold for myofibroblasts. To address this hypothesis, we analyzed BMs during alveologenesis by visualizing Collagen Type IV Alpha 1 (Col4a1), a major BM component. Col4a1 was distributed throughout the alveoli in both control and *Rap1iECKO* mice (Fig. 3a and Supplementary Fig. 3b). In controls, Col4a1 was well assembled and formed sheet-like BMs, which closely surrounded the alveolar myofibroblasts (Fig. 3a). In clear contrast, Col4a1 resulted in the construction of disorganized BMs that were loosely attached to the alveolar myofibroblasts in *Rap1iECKO* mice (Fig. 3a). Consistently, the single slice images revealed that control mice showed continuous BMs tightly attached to the alveolar myofibroblasts, while *Rap1iECKO* mice exhibited a discontinuous appearance of BMs surrounding the myofibroblasts (Fig. 3b–d and Supplementary Fig. 3c). We further investigated whether alveolar ECs in *Rap1iECKO* mice fail to form continuous BMs by conducting mosaic gene deletion analysis. To achieve mosaic deletion of *Rap1a/b* in ECs, a low dose of tamoxifen was delivered from the lactating mother into *Rap1afl/fl;Rap1bfl/fl;Cdh5-CreERT2* postnatal mice with the *mTmG* reporter background (*Rap1iECKO;mTmG* mice) at P1, enabling the identification of ECs lacking *Rap1a/b* with membrane GFP (mGFP) fluorescence (Supplementary Fig. 3d). *Rap1afl/+;Rap1bfl/+;Cdh5-CreERT2* or *Rap1afl/fl;Rap1bfl/fl;Cdh5-CreERT2* mice with the *mTmG* reporter background (*Rap1iECHet;mTmG* mice) served as a control. Col4a1 located around mGFP-labeled *Rap1a/b*-deficient ECs showed less assembly, forming discontinuous BMs, whereas that around the mGFP-labeled control ECs constructed continuous BMs (Supplementary Fig. 3e, f). These results raise the possibility of defective assembly of Col4a1 into stable BMs in *Rap1iECKO* mice. Additionally, we observed lower Col4a1 protein levels in the lungs of *Rap1iECKO* mice than in control lungs, despite similar mRNA levels of *Col4a1* and *Col4a2* in these two groups (Fig. 3e–g). Given that matrix metalloproteinase-2, also known as 72 kDa type IV collagenase, is known to be active during alveologenesis[37], it is reasonable to assume that Col4a1 in the alveoli of *Rap1iECKO* mice undergoes digestion by collagenolytic enzymes, possibly due to impaired Col4a1 assembly. We also analyzed laminin, another BM component, and found that its localization in BMs was relatively well preserved even in *Rap1iECKO* mice (Fig. 3b, c and Supplementary Fig. 3b, c), revealing that ECs recruit Col-4, but not laminin, into BMs in a Rap1-dependent manner.

Alveolar myofibroblasts secrete and assemble elastin into elastic fibers which are mechanically required for alveolar formation[9]. Thus, we next analyzed elastic fibers in alveoli and found that they localized around the alveolar entry ring in both control and *Rap1iECKO* mice (Fig. 3h). However, quantitative analysis revealed that the elastic fibers surrounding the alveolar entry ring were thinner in *Rap1iECKO* mice than in control mice, although expression of *Elastin* mRNA in the lungs differed minimally between these two groups (Fig. 3i, j and Supplementary Movie 3, 4), supporting the hypothesis that functions of alveolar myofibroblasts are impaired in *Rap1iECKO* mice. Collectively, these findings suggest that ECs Rap1-dependently generate BMs

required for myofibroblasts to induce mechanical signal-mediated alveologenesis (Fig. 3k).

## Rap1 stimulates adhesive activity of integrin β1 in alveolar ECs

We next investigated the molecular mechanism by which endothelial Rap1 induces BM formation. Rap1 stimulates the adhesive activity of integrins to promote cell adhesion to the extracellular matrix[38,39]. Importantly, a previous report showed that RAP-3, a *C. elegans* orthologue of mammalian Rap1, activates PAT-2 (α-subunit)/PAT-3 (β-subunit) integrins, which in turn promote Col-4 recruitment into BMs in a laminin-independent manner[40]. Based on these findings, we hypothesized that endothelial Rap1 activates integrins to induce Col-4 recruitment to BMs. To test this hypothesis, we first evaluated the adhesive activity of integrins in ECs of control and *Rap1iECKO* mouse lungs. To this end, ECs were isolated from the lungs of control and *Rap1iECKO* mice, plated onto dishes coated with Collagen Type I (Col-1), and subjected to analysis of integrin-mediated cell adhesion. *Rap1iECKO* mouse-derived ECs exhibited a significantly smaller cell area than control ECs (Fig. 4a, b). In addition, the number and size of focal adhesions (FAs) and focal complexes (FCs) were reduced in the ECs derived from *Rap1iECKO* mice as compared to those of control ECs (Fig. 4a, c), suggesting decreased adhesive activity of integrins in *Rap1*-deficient lung ECs. Consistently, the number of activated integrin β1-positive FAs/FCs was significantly decreased in the lung ECs isolated from *Rap1iECKO* mice as compared to those from control mice (Fig. 4d, e). These findings suggest that Rap1 is required for activation of integrins in lung ECs.

To further confirm the Rap1-mediated activation of integrins in alveolar ECs, we performed 3D immunofluorescence analysis of the lungs of *Rap1iECHet;mTmG* and *Rap1iECKO;mTmG* mice with the antibodies against integrin β1and its activated form. In both groups, integrin β1 was localized at the mGFP-labeled plasma membrane of alveolar ECs (Fig. 4f). However, the plasma membrane localization of the activated form of integrin β1 was significantly diminished in alveolar ECs of *Rap1iECKO;mTmG* mice as compared to those of *Rap1iECHet;mTmG* mice (Fig. 4f, g), indicating that Rap1 is required for activation of integrins in alveolar ECs. Furthermore, we observed that Col4a1 was assembled near the plasma membrane-localized activated integrin β1 in alveolar ECs of control mice (Fig. 4h). In contrast, alveolar ECs of *Rap1iECKO* mice exhibited a reduced plasma membrane localization of activated integrin β1 and Col4a1 assembly was only slightly induced around these ECs (Fig. 4h). Considered collectively, our results suggest that Rap1 activates integrin β1 in alveolar ECs to induce BM formation.

## Rap1 generates BMs through integrin-mediated recruitment of Col-4

We next aimed to elucidate whether Rap1 induces Col-4 recruitment by activating integrin in ECs. For this purpose, lung ECs isolated from control and *Rap1iECKO* mice were cultured in Col-1-coated dishes for 72 h and then subjected to immunofluorescence staining with Col4a1 antibody. Col4a1 was deposited and assembled on the surface of the dish by control lung ECs (Fig. 5a, b and Supplementary Fig. 4a). However, the amount of Col4a1 assembled on the dish was significantly decreased when lung ECs derived from *Rap1iECKO* mice were cultured (Fig. 5a, b and Supplementary Fig. 4a). We also conducted the same experiments using human umbilical vein endothelial cells (HUVECs). Control HUVECs deposited and assembled COL4A1 on the culture dish surface (Fig. 5c, d). However, depletion of both *RAP1A* and *RAP1B* led to a reduction of COL4A1 assembled on the dish without affecting the mRNA levels of *COL4A1* and *COL4A2* (Fig. 5c, d and Supplementary Fig. 4b–f). We also tested the possibility that the decreased amount of Col-4 assembled on the dish cultured with *Rap1*-depleted ECs is due to its degradation by performing Western blot analysis and found that the cellular protein levels of COL4A1 differed minimally between control and *RAP1A/RAP1B*-depleted HUVECs (Fig. 5e, f). In addition, similar

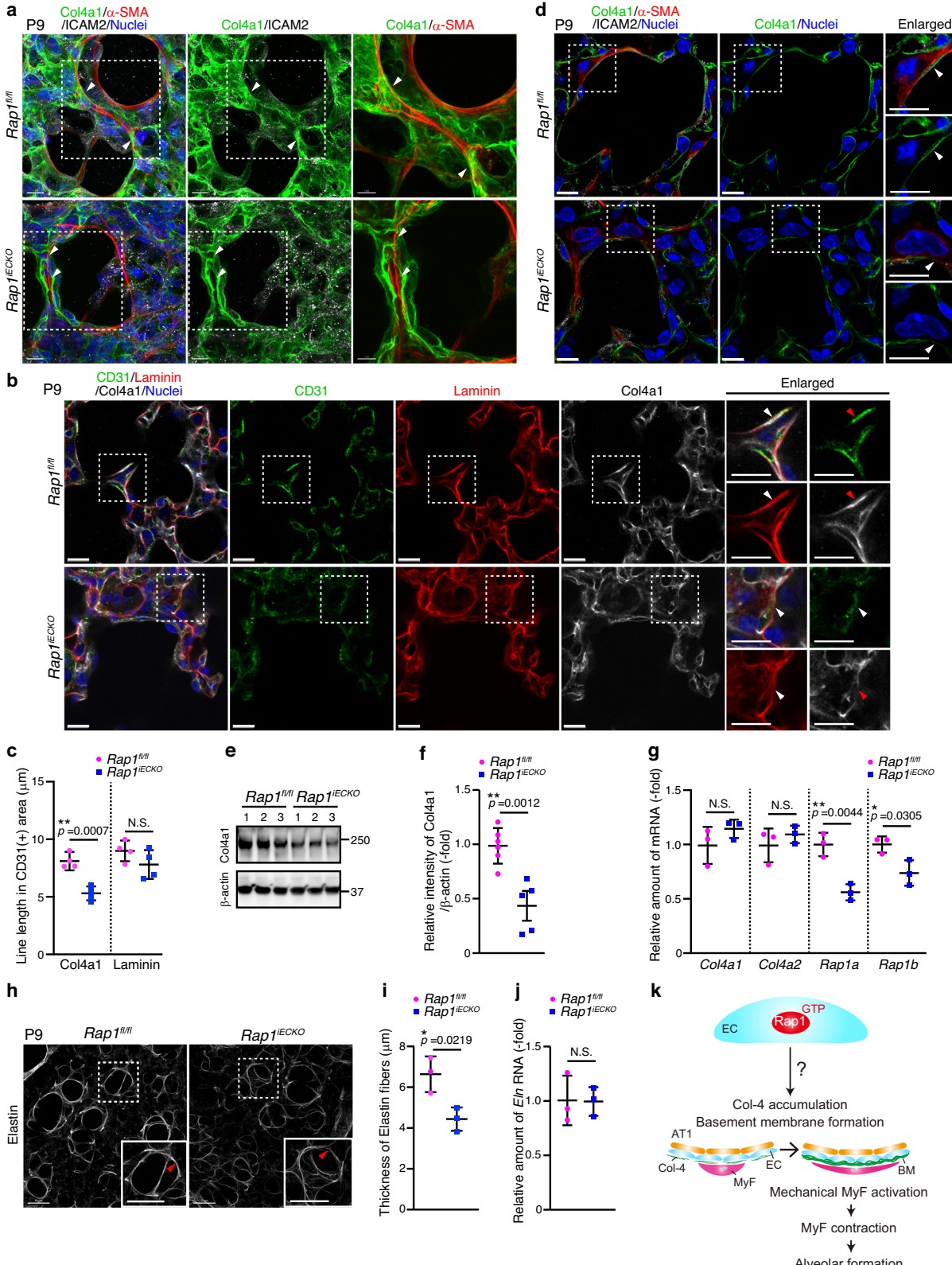

amounts of COL4A1 protein were present in the conditioned media cultured with control and *RAP1A/RAP1B*-depleted HUVECs (Fig. 5e, f). These findings indicate that the depletion of Rap1 does not affect either the degradation or the secretion of Col-4 and thereby suggest that endothelial Rap1 actively regulates the assembly of Col-4.

Next, we investigated whether Rap1 induces Col-4 assembly through the activation of integrins. Manganese ions (Mn$^{2+}$) reportedly

increase the ligand-binding affinity of integrins[41]. Indeed, the treatment of HUVECs with Mn$^{2+}$ increased the active form of integrin β1 localized at FAs (Supplementary Fig. 4g). Thus, we examined the effect of Mn$^{2+}$ treatment on COL-4 assembly and showed that it partially rescued the assembly of COL4A1 by *RAP1A/RAP1B*-depleted HUVECs (Fig. 5c, d). These results suggest that ECs deposit and assemble Col-4 through Rap1-mediated activation of integrins.

**Fig. 3 | ECs induce Col-4 accumulation to generate BMs acting as a scaffold for myofibroblasts in a Rap1-dependent manner. a** Confocal z-projection images for Col4a1 (green), α-SMA (red), ICAM2 (white), and Nuclei (DAPI, blue) in alveoli from *Rap1^(fl/fl)* and *Rap1^(iECKO)* mice at P9. Arrowheads, Col4a1-based BMs surrounding myofibroblasts. **b** Single slice images for CD31 (green), Laminin (red), Col4a1 (white), and Nuclei (DAPI, blue) in alveoli from *Rap1^(fl/fl)* and *Rap1^(iECKO)* mice at P9. Arrowheads, Col4a1 and Laminin associated with ECs. **c** Line lengths of Col4a1 (left) and Laminin (right) in CD31(+) areas, as in (**b**) (*n* = 4 mice/each). **d** Single slice images for Col4a1 (green), α-SMA (red), ICAM2 (white), and Nuclei (DAPI, blue) in alveoli from *Rap1^(fl/fl)* and *Rap1^(iECKO)* mice at P9. Arrowheads, BMs localizing between ECs and myofibroblasts. **e** Western blot analysis of expressions of Col4a1 and β-actin in lungs of *Rap1^(fl/fl)* and *Rap1^(iECKO)* mice at P9 (3 mice/each). **f** Expression levels of Col4a1 relative to those of β-actin, as in (**e**) (*Rap1^(f/f)* and *Rap1^(iECKO)*, *n* = 6 and 5 mice).

**g** Expression levels of *Col4a1*, *Col4a2*, *Rap1a*, and *Rap1b* mRNA relative to those of *Actb* in lungs of *Rap1^(fl/fl)* and *Rap1^(iECKO)* mice at P9 (*n* = 3 mice/each). **h** Confocal z-projection images for Elastin in alveoli from *Rap1^(fl/fl)* and *Rap1^(iECKO)* mice at P9. **i** Thicknesses of Elastin fibers, as in (**h**) (*n* = 3 mice/each). **j** Expression levels of *Eln* mRNA relative to those of *Actb* in lungs of *Rap1^(fl/fl)* and *Rap1^(iECKO)* mice at P9 (*n* = 3 mice/each). **k** Schematic diagram illustrating how ECs regulate the function of alveolar myofibroblasts. ECs induce BM formation by accumulating Col-4 in a Rap1-dependent manner. BMs constructed by ECs appear to serve as a scaffold for myofibroblasts to induce alveolar formation. Each dot represents an individual mouse, and data are presented as means ± s.d. (**c**, **f**, **g**, **i**, **j**). N.S. no significance; **P < 0.01, *P < 0.05, by two-tailed Student's *t*-test (**c**, **f**, **g**, **i**, **j**). Boxed areas are enlarged on the right (**a**, **d**, **b**) and in the insets (**h**). Scale bars; 10 µm (**a**, **d**), 7 µm (enlarged in **a**), 15 µm (**b**), 50 µm (**h**). Source data are provided as a Source data file.

To further confirm this notion, we performed a Matrigel-based in vitro cord formation assay using HUVECs. HUVECs were plated on Matrigel, allowed to form capillary-like network structures for 36 h, and then stained with anti-COL4A1 antibody to analyze BM formation. COL4A1 was accumulated around HUVEC-constructed cord-like structures to form BMs (Fig. 5g, h), a process completely inhibited by knockdown of *COL4A1*, indicating that COL4A1 assembled on the vessel wall had been deposited by HUVECs (Supplementary Fig. 4h–j). Depletion of both *RAP1A* and *RAP1B* severely inhibited accumulation of COL4A1 around the cord-like structures, while not impacting the expression levels of *COL4A1* and *COL4A2* mRNAs in HUVECs (Fig. 5g, h, Supplementary Fig. 4d). Furthermore, the $Mn^{2+}$ treatment partially, yet significantly, rescued the diminished accumulation of COL4A1 around the cord-like structures in HUVECs depleted of both *RAP1A* and *RAP1B* (Fig. 5g, h). To further confirm the role of integrin β1 in BM formation, we treated HUVECs with blocking antibody against integrin β1 (mAb13)[42], and found that COL4A1 accumulation around the cord-like structures was decreased by integrin β1 blocking antibody (Supplementary Fig. 4k, l). Taken together, these findings indicate that Rap1 induces Col-4 accumulation around vessel walls to form BM by stimulating the adhesive activity of integrins.

### Integrin β1 generates BMs required for myofibroblast-mediated alveologenesis

Based on the above findings, we tested whether integrin β1 acts downstream from Rap1 to form the BMs required for mechanical signaling in myofibroblasts. To this end, we generated EC-specific *Itgb1* deficient (*Itgb1^(iECKO)*) mice by crossing floxed *Itgb1* mice with *Cdh5-CreERT2* transgenic mice. Endothelial *Itgb1* was deleted after birth by delivering tamoxifen via the lactating mother for three consecutive days from P0. Expression of *Itgb1* mRNA was partially, but significantly, decreased in CD31(+) ECs isolated from P9 lungs of *Itgb1^(iECKO)* mice as compared to those from control mice, while its expressions in PDGFRα(+) cells containing myofibroblasts were similar in control and *Itgb1^(iECKO)* mice (Supplementary Fig. 5a, b).

Therefore, we investigated the requirement of endothelial integrin β1 for alveolar morphogenesis. As observed in *Rap1^(iECKO)* mice (Fig. 1d, e), *Itgb1^(iECKO)* mice showed an increased MLI compared to control mice, indicating compromised alveologenesis in the *Itgb1^(iECKO)* mice (Fig. 6a, b). Thus, we next tested whether the mechanical signaling required for alveologenesis is impaired in myofibroblasts of *Itgb1^(iECKO)* mice. Although myofibroblasts localizing to the alveolar entry ring were present in *Itgb1^(iECKO)* mice, their α-SMA filaments were thinner than those of control mice, as observed in *Rap1^(iECKO)* mice (Figs. 2c, d, and 6c). Consistently, the pMLC signal in alveolar myofibroblasts of *Itgb1^(iECKO)* mice was dramatically attenuated compared to that in control siblings (Fig. 6d, e). Furthermore, fewer alveolar myofibroblasts exhibited the nuclear YAP signal in *Itgb1^(iECKO)* mice than in control mice (Fig. 6f, g). These results show that mechanical signaling in alveolar myofibroblasts was compromised not only in *Rap1^(iECKO)* mice but also in *Itgb1^(iECKO)* mice.

Finally, we investigated whether the BM formation required for mechanical signaling in myofibroblasts was inhibited in *Itgb1^(iECKO)* mice. In control mice, BMs constructed by well-assembled Col4a1 tightly surrounded myofibroblasts localizing to the alveolar entry ring (Fig. 7a). In contrast, BMs consisting of Col4a1 showed disorganized morphology and were loosely attached to the alveolar myofibroblasts in *Itgb1^(iECKO)* mice (Fig. 7a). Single slice images of alveoli also revealed that linear and continuous BMs constructed by Col4a1 and laminin were established along the ECs in control mice, whereas both Col4a1 and laminin were diffusely and discontinuously localized around the ECs in the alveoli of *Itgb1^(iECKO)* mice (Fig. 7b, c and Supplementary Fig. 6). Altogether, these results indicate that endothelial Rap1 stimulates integrin β1 to induce the formation of BMs, which act as a scaffold for myofibroblasts and thereby induce mechanical signal-mediated alveologenesis (Fig. 8).

### Discussion

In this study, we uncovered a molecular mechanism underlying EC-mediated alveolar morphogenesis by analyzing EC-specific *Rap1*-deficient mice (Fig. 8). We demonstrated that *Rap1^(iECKO)* mice exhibit hypo-alveolarization due to defective mechanical signaling in alveolar myofibroblasts, which is essential for the formation of secondary septa. As to the underlying mechanism, we discovered that alveolar ECs accomplish integrin β1-mediated recruitment of Col-4 and BM assembly in a Rap1-dependent manner, thereby acting as a scaffold for myofibroblasts allowing mechanical signal-induced secondary septa formation during alveologenesis.

Hypo-alveolarization in the lungs of *Rap1^(iECKO)* mice is not secondary to growth retardation and/or defects in the development of other organs. It has been suggested that growth retardation and calorie restriction disrupt proper alveologenesis in postnatal and adult mice[43,44]. These findings raise the possibility that impaired formation of organs other than the lungs and subsequent growth retardation may cause hypo-alveolarization in *Rap1^(iECKO)* mice, since *Rap1* was deleted in all ECs throughout the body. However, this is not the case, since *Rap1^(iECKO)* postnatal mice did not exhibit growth retardation, at least until P14, and the liver, stomach, and intestine developed normally. Consistent with these observations, mosaic gene deletion analysis of *Rap1* in ECs further confirmed the requirement of *Rap1* in alveolar ECs for proper alveologenesis. Thus, the growth retardation and defective formation of organs other than the lungs do not cause hypo-alveolarization in *Rap1^(iECKO)* postnatal mice.

ECs contribute to organ morphogenesis not only by producing angiocrine factors but also via their cellular functions. It is now widely accepted that ECs produce angiocrine factors that actively regulate the development of various organs such as the lungs[15], heart[45], kidneys[46], liver[47], pancreas[48], and bone[49]. Our present study revealed a mode of EC-mediated organ development in which ECs regulate alveolar morphogenesis via the formation of BMs. Similarly, we recently reported that ECs promote glomerular morphogenesis possibly by inducing blood filtration during pronephros formation in zebrafish embryos[50].

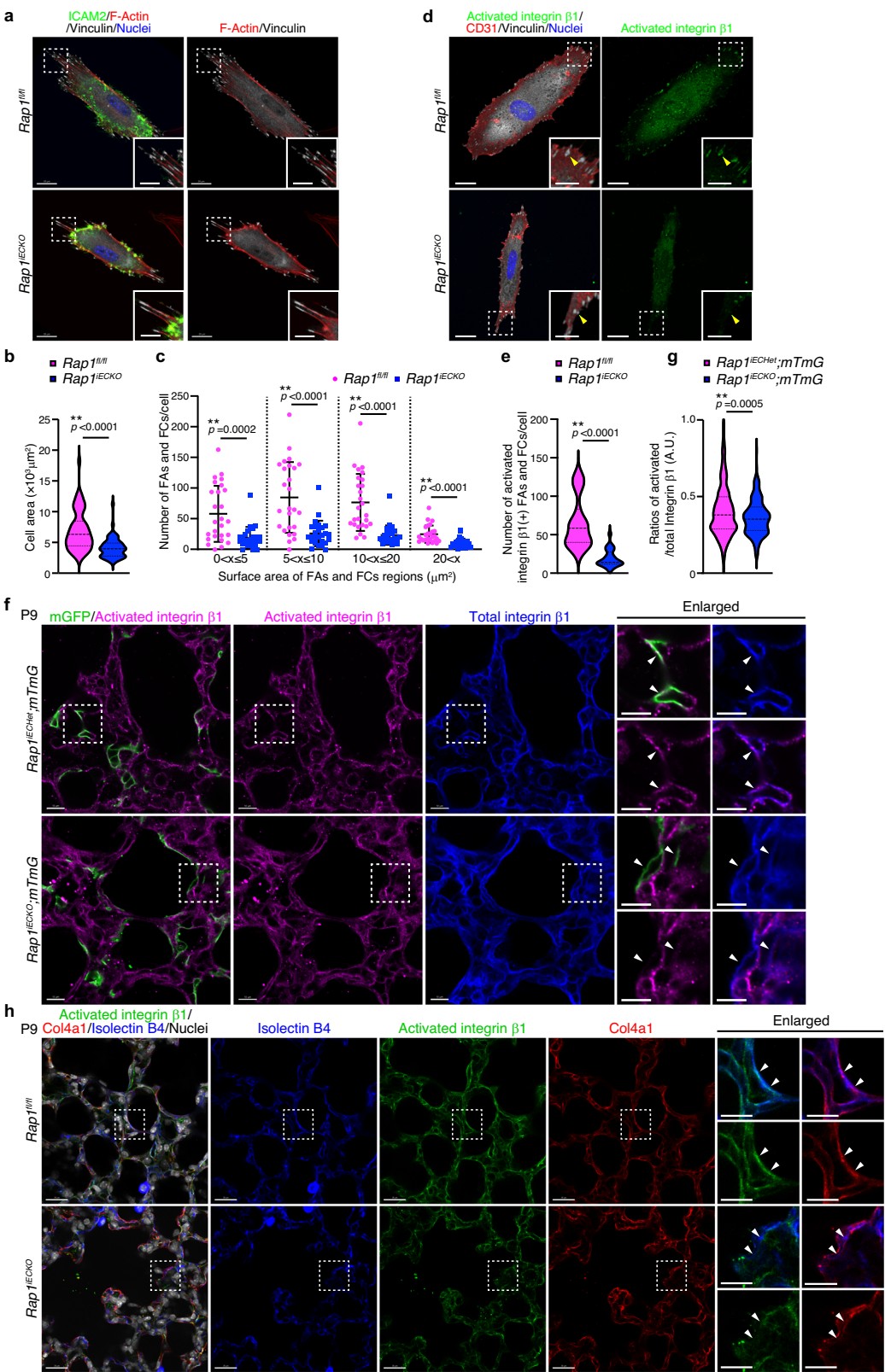

Hence, cellular functions mediated by ECs might play a crucial role in various types of organ morphogenesis. In addition, such intrinsic EC functions might also regulate organ regeneration and the pathogenesis of various diseases, as do EC-derived angiocrine factors. Thus, future studies are anticipated to clarify further unexpected roles of EC functions, deepening our understanding of the regulatory mechanisms of organ formation, maintenance, regeneration and dysfunction.

How do ECs induce Col-4 assembly to form BMs through Rap1-mediated activation of integrin β1? BMs are composed of laminin, Col-4, nidogen, and perlecan, among which laminin and Col-4 self-assemble into two independent supramolecular networks[51,52]. BM formation reportedly begins through laminin binding to cell surface receptors such as integrins and dystroglycan[52]. Laminin scaffolds on the cell surface subsequently recruit Col-4 networks via nidogens. Thus, Col-4

**Fig. 4 | Rap1 stimulates adhesive activity of integrin β1 in alveolar ECs.**
**a**–**c** Formation of FAs and FCs in ECs isolated from the lungs of P9 *Rap1*<sup>fl/fl</sup> and *Rap1*<sup>iECKO</sup> mice and cultured on collagen-coated dishes for 72 h. **a** Confocal fluorescence images for ICAM2 (green), F-actin (red), vinculin (white), and Nuclei (DAPI, blue). **b** Violin plots of the average cell area (*Rap1*<sup>fl/fl</sup> and *Rap1*<sup>iECKO</sup>, n = 90 and 71 cells from 3 mice). **c** Number of FAs/FCs per cell, within the indicated size range at the bottom. The dots represent individual cells. Data are means ± s.d. (*Rap1*<sup>fl/fl</sup> and *Rap1*<sup>iECKO</sup>, n = 90 and 71 cells from 3 mice). **d** Confocal fluorescence images for activated integrin β1 (green), CD31 (red), vinculin (white), and Nuclei (DAPI, blue) in ECs isolated from the lungs of P9 *Rap1*<sup>fl/fl</sup> and *Rap1*<sup>iECKO</sup> mice and cultured for 72 h. Arrowheads, the activated integrin β1-positive FAs/FCs. **e** Violin plots indicate the average number of activated integrin β1-positive FAs/FCs per cell, as in **d** (*Rap1*<sup>fl/fl</sup> and *Rap1*<sup>iECKO</sup>, n = 23 and 24 cells from 3 mice). **f** Single slice images for mGFP (green), activated integrin β1 (magenta), and integrin β1 (blue) in alveoli from *Rap1*<sup>iECHet</sup>;*mTmG* and *Rap1*<sup>iECKO</sup>;*mTmG* mice at P9. Arrowheads, mGFP-labeled plasma membranes of ECs. **g** Violin plots showing the ratio of the number of activated integrin β1 to the total integrin β1 per cell (arbitrary units, A.U.), as in (**f**) (*Rap1*<sup>iECHet</sup>;*mTmG* and *Rap1*<sup>iECKO</sup>;*mTmG*, n = 433 and 217 ROIs from 3 mice). **h** Single slice images for activated integrin β1 (green), Col4a1 (red), isolectin B4 (blue), and Nuclei (DAPI, white) in alveoli from *Rap1*<sup>fl/fl</sup> and *Rap1*<sup>iECKO</sup> mice at P9. Arrowheads, isolectin B4(+) ECs. In violin plots (**b**, **e**, **g**) bold and thin dashed lines indicate the median and quartiles, respectively. **P < 0.01, by two-tailed Student's t-test (**b**, **c**, **e**, **g**). Boxed areas are enlarged in the insets (**a**, **d**) and on the right (**f**, **h**). Scale bars; 20 μm (**a**, **d**, **h**), 10 μm (**f** and enlarged images in **a**, **d**, **h**) and 7 μm (enlarged images in **f**). Source data are provided as a Source data file.

incorporation into BMs is thought to require laminin scaffolds on cell surfaces. However, in *Rap1*<sup>iECKO</sup> mice, Col-4 localization in BMs around alveolar myofibroblasts was disorganized, despite laminin localization being relatively well preserved, indicating that laminin scaffolds are not sufficient to correctly accumulate Col-4 at BMs. In this regard, laminin-independent mechanisms for recruiting Col-4 into BMs have been reported[40,53]. Jayadev et al. showed that Col-4 is incorporated into the gonadal and pharyngeal BMs in *C. elegans* at a postembryonic stage in laminin-dependent and laminin-independent manners, respectively. They showed that INA-1/PAT-3 integrin was selectively activated to recruit laminin at the gonad, thereby directing Col-4 incorporation into BMs, whereas PAT-2/PAT-3 integrin promoted Col-4 recruitment into the pharyngeal BMs in a laminin-independent manner. Importantly, they also found that Rap1 activates the PAT-2/PAT-3 integrins to recruit Col-4. Therefore, ECs might stimulate Rap1 to activate collagen-binding integrins, which would in turn recruit Col-4 into BMs required for myofibroblast function. Among the collagen-binding integrins, α1β1 and α2β1 integrins are known to be major receptors for Col-4 and to regulate diverse biological functions such as cell adhesion, cell migration, insulin secretion, and inflammation by binding to Col-4[54,55]. Indeed, publicly available scRNA-seq data of the lungs showed that ECs in lungs of postnatal mice express *Itga1* and *Itga2* as well as *Itgb1*[56]. Thus, α1β1 and α2β1 integrins might be activated by Rap1 in ECs, thereby binding and accumulating Col-4 to generate BMs. On the other hand, *Itgb1*<sup>iECKO</sup> mice exhibited impaired accumulation of both Col-4 and laminin at BMs. Hence, ECs might also promote laminin incorporation into BMs by stimulating laminin-binding integrins such as α6β1 integrins independently of Rap1.

EC-generated BMs act as a stiff scaffold for alveolar myofibroblasts, leading to the induction of alveologenesis through activation of mechanical signaling. Substrate stiffness influences a wide range of cellular responses such as cell adhesion, migration, survival, proliferation, and differentiation[36,57]. Indeed, differentiation of fibroblasts into myofibroblasts and their cellular functions are known to be markedly affected by matrix stiffness[58]. During alveologenesis, myofibroblasts contract to form secondary septa by activating mechanical signaling involving myosin activation and nuclear YAP localization. Activation of such mechanical signaling is thought to depend on substrate stiffness[59]. Therefore, it is assumed that alveolar myofibroblasts induce activation of mechanical signaling by sensing the stiffness of EC-generated BMs. Indeed, BM mechanics have been widely suggested to regulate development, homeostasis, and various disease processes[33,60]. If this is the case, how do myofibroblasts sense the stiffness of EC-generated BMs? Cells sense substrate stiffness via integrin-mediated cell adhesion complexes, which link the extracellular matrix to the actin cytoskeleton[61]. Therefore, myofibroblasts might sense BM stiffness by adhering to BM components such as Col-4 via integrins. In addition, a prior study showed that BMs promote rapid and robust fibronectin assembly using sliding FAs driven by a contractile winch, thereby leading to accumulation of fibronectin at BMs[62]. It has also been reported that alveolar myofibroblasts adhere to fibronectin via α5β1 integrins, which is a requirement for alveologenesis[63]. Thus, myofibroblasts might sense the stiffness of BMs through integrin-mediated adhesion to fibronectin assembled on the EC-generated BMs. These hypotheses merit detailed examination in future studies.

This study has limitations, one of which is that the cause of lethality in *Rap1*<sup>iECKO</sup> mice was not identified. The hypo-alveolarization exhibited by *Rap1*<sup>iECKO</sup> mice might not be the cause of postnatal death, since suppression of alveolar septation reportedly does not lead to postnatal death[64]. Hence, further investigation is required to identify the cause of death in *Rap1*<sup>iECKO</sup> mice. Another limitation is that our results do not reveal the morphogenetic processes of alveolar formation. We analyzed alveologenesis by performing 3D immunofluorescence analysis of fixed lung tissues. However, analyzing 3D still images of alveoli is insufficient to illustrate morphogenetic processes of alveolar formation, which are highly complex and dynamic. Thus, live-imaging techniques, which are challenging to perform, are required to meaningfully elucidate the morphogenetic processes of alveolar formation.

In summary, our data demonstrate that ECs facilitate alveologenesis in postnatal lungs by constructing BMs through Rap1-mediated activation of integrin β1, which is required for activation of mechanical signaling in myofibroblasts. Our findings are not only important for understanding lung development but may also have implications relevant to lung disease. Surgical removal of lung lobes, i.e., pneumonectomy, induces compensatory growth of the remaining lobes not only through the growth of existing alveoli but also via the formation of new alveolar units by reactivated alveolar myofibroblasts[65,66]. Stimulating EC-mediated BM formation might be an effective therapeutic strategy to induce realveolarization, not only after pneumectomy, but also for lung diseases such as bronchopulmonary dysplasia and chronic obstructive pulmonary disease.

## Methods
### Mice
Animal experiments were approved by the animal committees of the National Cerebral and Cardiovascular Center, the Nippon Medical School, and Tokyo Medical and Dental University and performed by following the guidelines of the National Cerebral, Cardiovascular Center and the Nippon Medical School, and Tokyo Medical and Dental University.

Mice were housed in 12:12 light:dark light cycles at ambient temperature ranging between 20 °C and 23 °C and humidities between 30% and 70%. All mouse lines used in this study were in the C57BL/6 J background. *Rap1a* and *Rap1b* double-floxed mice (B6;129S-*Rap1a*<sup>tmMorz</sup>;*Rap1b*<sup>tm1Morz</sup>/J) and *Gt(ROSA)26Sor*<sup>tm4(ACTB-tdTomato,-EGFP)Luo</sup>/J, a cell membrane-targeted, two-color fluorescent Cre-reporter line (*ROSA-mTmG*) were obtained from The Jackson Laboratory[67]. *Cdh5-CreERT2* mice expressing CreERT2 recombinase under the control of the *Cdh5* promoter[68] and *Itgb1*-floxed mice[69] were previously described in detail.

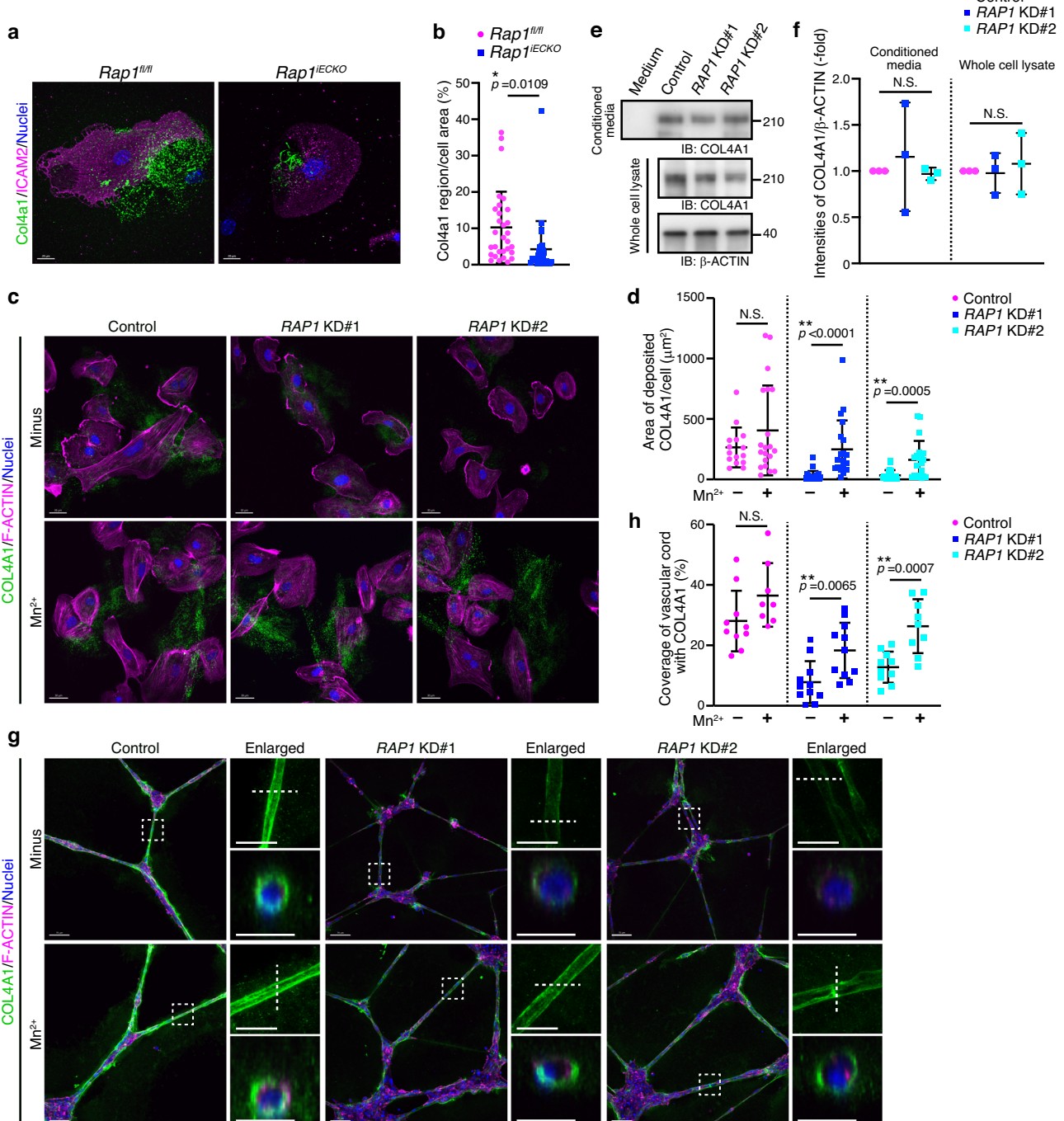

**Fig. 5 | Endothelial Rap1 induces integrin-mediated recruitment of Col-4 to generate BMs. a** Confocal fluorescence images for Col4a1 (green), ICAM2 (magenta), and Nuclei (DAPI, blue) in ECs from the lungs of P9 *Rap1^fl/fl* and *Rap1^iECKO* mice and cultured on collagen-coated dishes for 72 h. **b** The amounts of Col4a1 assembled on the dish, as in (**a**), are expressed as percentages of Col4a1-positive area relative to ICAM2-labeled total cellular area (*Rap1^f/f* and *Rap1^iECKO*, *n* = 39 and 44 cells from 3 mice). **c** Confocal fluorescence images for COL4A1 (green), F-Actin (magenta), and Nuclei (DAPI, blue) in HUVECs transfected with control siRNA or two siRNA mixtures targeting both *RAP1A* and *RAP1B* (*RAP1* KD#1, *RAP1* KD#2) and cultured without or with 0.5 mM MnCl₂ for 24 h. **d** The amounts of COL4A1 assembled on the dish, as in (**c**), are expressed as a COL4A1-positive area divided by the number of cells in each image (Numbers of analyzed cells are provided in Supplementary Table 3). **e** Western blot analysis of COL4A1 and β-actin. Upper, COL4A1 in the conditioned media cultured without (Medium) or with HUVECs transfected with the indicated siRNAs; lower, COL4A1 and β-actin in the corresponding whole cell lysates. **f** The amounts of COL4A1 in the conditioned media and whole cell lysates normalized by those of β-actin, as in (**e**), are expressed relative to the levels observed in the control groups (*n* = 3 independent experiments). **g** Confocal z-projection images for COL4A1 (green), F-Actin (magenta), and Nuclei (DAPI, blue) in vascular cord structures constructed by HUVECs transfected with the indicated siRNAs and cultured on Matrigel without or with 0.5 mM MnCl₂ for 24 h. Boxed areas are enlarged on the upper right. The cross-sectional images of the areas indicated by dotted lines are shown at the bottom. **h** Coverage of vascular cord structures with COL4A1, as in (**g**), are expressed as percentages relative to total vascular cord areas (Numbers of experiments are provided in Supplementary Table 3). Each dot represents an individual cell (**b**), image (**d**, **h**), and experiment (**f**). Data are presented as means ± s.d. (**b**, **d**, **f**, **h**). N.S. no significance; **P < 0.01, *P < 0.05, by two-tailed Student's *t*-test (**b**, **d**, **h**) and by one-way ANOVA followed by Turkey's post-hoc test (**f**). Scale bars; 20 μm (**a**), 30 μm (**c** and upper enlarged images in **g**), 70 μm (**g**), 15 μm (lower enlarged images in **g**). Source data are provided as a Source data file.

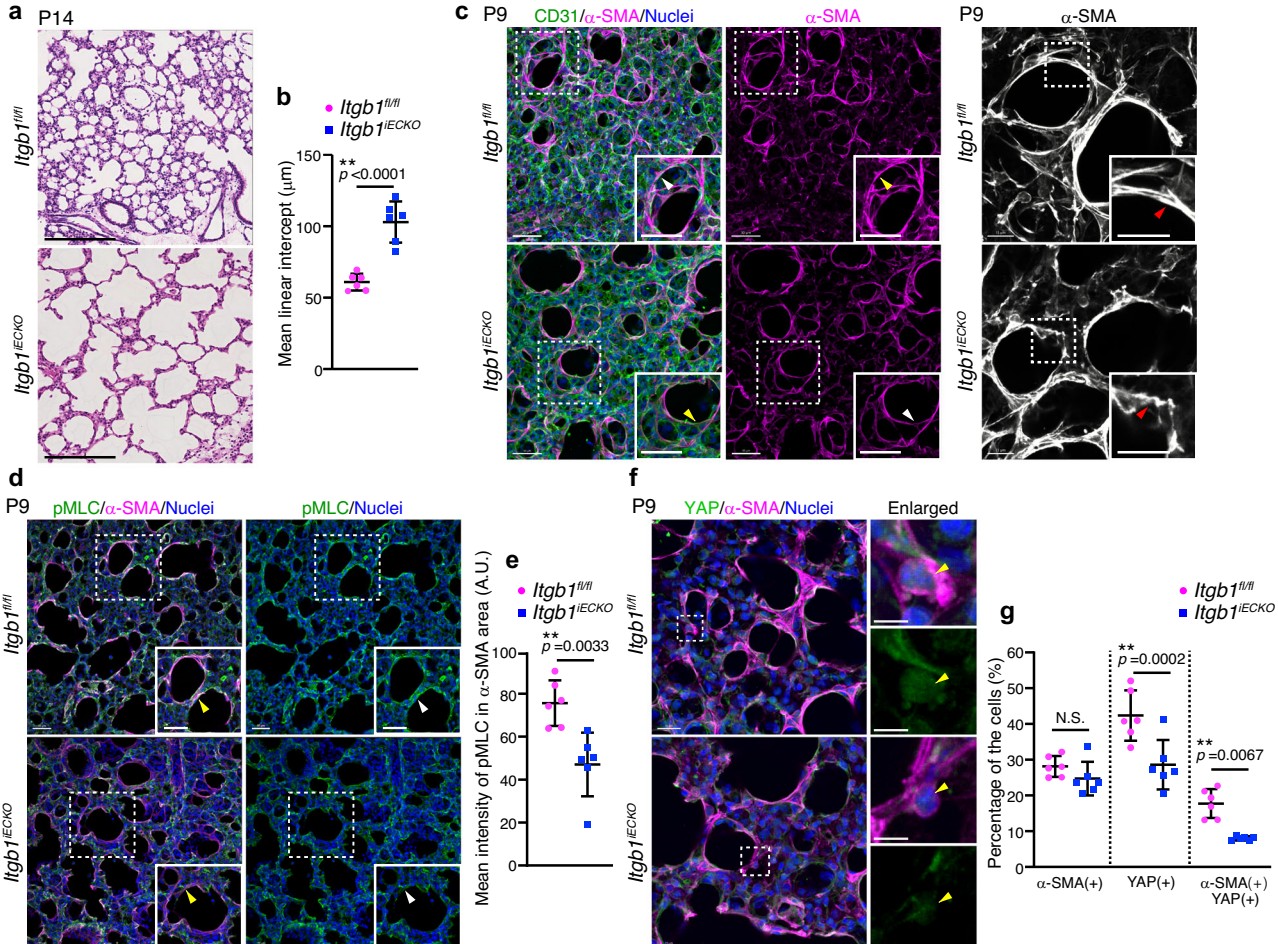

**Fig. 6 | Endothelial integrin β1 regulates alveologenesis by stimulating mechanical signaling in myofibroblasts. a** Images of HE-stained lung tissues from *Itgb1^{fl/fl}* and *Itgb1^{iECKO}* mice at P14. **b** Mean linear intercepts (MLI) in the lungs of *Itgb1^{fl/fl}* and *Itgb1^{iECKO}* mice at P14, as in (**a**) (*n* = 6 mice/each). **c** Confocal z-projection images for CD31 (green), α-SMA (magenta), and Nuclei (DAPI, blue) (left two columns) and for α-SMA (white) (right column) in alveoli from *Itgb1^{fl/fl}* and *Itgb1^{iECKO}* mice at P9. Boxed areas are enlarged in the insets. Arrowheads, α-SMA filaments surrounding the alveolar entry ring. **d** Confocal z-projection images for pMLC (green), α-SMA (magenta), and Nuclei (DAPI, blue) in alveoli from *Itgb1^{fl/fl}* and *Itgb1^{iECKO}* mice at P9. Boxed areas are enlarged in the insets. Arrowheads, the pMLC signal in myofibroblasts. **e** Fluorescence intensity of pMLC in α-SMA(+) areas

(arbitrary units, A.U.), as in (**d**) (*n* = 6 mice/each). **f** Confocal z-projection images for YAP (green), α-SMA (magenta), and Nuclei (DAPI, blue) in alveoli from *Itgb1^{fl/fl}* and *Itgb1^{iECKO}* mice at P9. Boxed areas are enlarged on the upper right. Enlarged YAP images are shown at the bottom. Arrowheads, the nuclei of α-SMA(+) myofibroblasts. **g** Percentages of α-SMA(+), nuclear YAP(+), and α-SMA(+)/nuclear YAP(+) cell populations, as in (**f**) (*n* = 6 mice/each). Each dot represents an individual mouse, and data are presented as means ± s.d. (**b, e, g**). N.S. no significance; **$P < 0.01$, by two-tailed Student's *t*-test (**b, e, g**). Scale bars; 50 μm (**a, d**, and left images in **c**), 20 μm (**f**), 15 μm (right images in **c**), and 10 μm (enlarged images in **f**). Source data are provided as a Source data file.

*Rap1a^{fl/fl};Rap1b^{fl/fl}* mice and *Itgb1^{fl/fl}* mice were crossed with *Cdh5-CreERT2* mice to specifically delete both *Rap1a* and *Rap1b* and *Itgb1* in ECs upon tamoxifen treatment. *Rap1a^{fl/fl};Rap1b^{fl/fl}* mice and *Itgb1^{fl/fl}* mice were used as littermate controls. To achieve tamoxifen-induced recombination in postnatal mice, 20 mg/ml tamoxifen (T5648, Sigma–Aldrich) dissolved in corn oil was orally administered into the lactating dams for three consecutive days from P0 (75 μg/g/day). Littermate control postnatal mice also received tamoxifen from the lactating dams. A 20 mg dose of tamoxifen was dissolved with 1 ml of corn oil by sonication followed by vigorous shaking at 65 °C under dark conditions, and then stored at −20 °C for up to 1 week in the dark.

The mosaic depletion of *Rap1a and Rap1b* was carried out as describe in the Supplementary Fig. 3d. The *Rap1a^{fl/fl};Rap1b^{fl/fl};Cdh5-CreERT2* mice were crossed with the *Rap1a^{fl/+};Rap1b^{fl/+};ROSA-mTmG* mice. Then, the mother was orally gavaged with amoxifen (7.5 μg/g/day) only at P1, administering a low dose of tamoxifen to the *Rap1a^{fl/fl};Rap1b^{fl/fl};Cdh5-CreERT2;ROSA-mTmG* (*Rap1^{iECKO};mTmG*) pups through the mother's milk. The alveolar ECs lacking the *Rap1a and Rap1b* were identified based

on the fluorescence of mGFP. The *Rap1a^{fl/+};Rap1b^{fl/+};Cdh5-CreERT2;ROSA-mTmG* mice or *Rap1a^{fl/fl};Rap1b^{fl/fl};Cdh5-CreERT2;ROSA-mTmG* (*Rap1^{iECHet};mTmG*) mice served as a control.

### HE staining

Lungs, liver, stomach, and small intestine were perfused with phosphate-buffered saline (PBS) through the right ventricle of the heart, then inflated by injecting 4% paraformaldehyde (PFA)/PBS containing $NaN_3$ (PBS/$NaN_3$) through the trachea, and fixed for 15 min. Then, the whole organs were harvested and kept overnight in 4% PFA/PBS/$NaN_3$ at 4 °C. After fixation, they were dehydrated and embedded in paraffin wax. The paraffin-embedded blocks of the organs were cut into 4 μm tissue sections and stained with a standard HE staining procedure using 1% Eosin Y Solution (Wako) and Mayer's Hematoxylin Solution (Wako) according to a previously described method[37]. Then, the specimens were mounted in Canada Balsam (KANTO CHEMICAL CO., INC.) and imaged employing fluorescence microscopy (KEYENCE, BZ-X710).

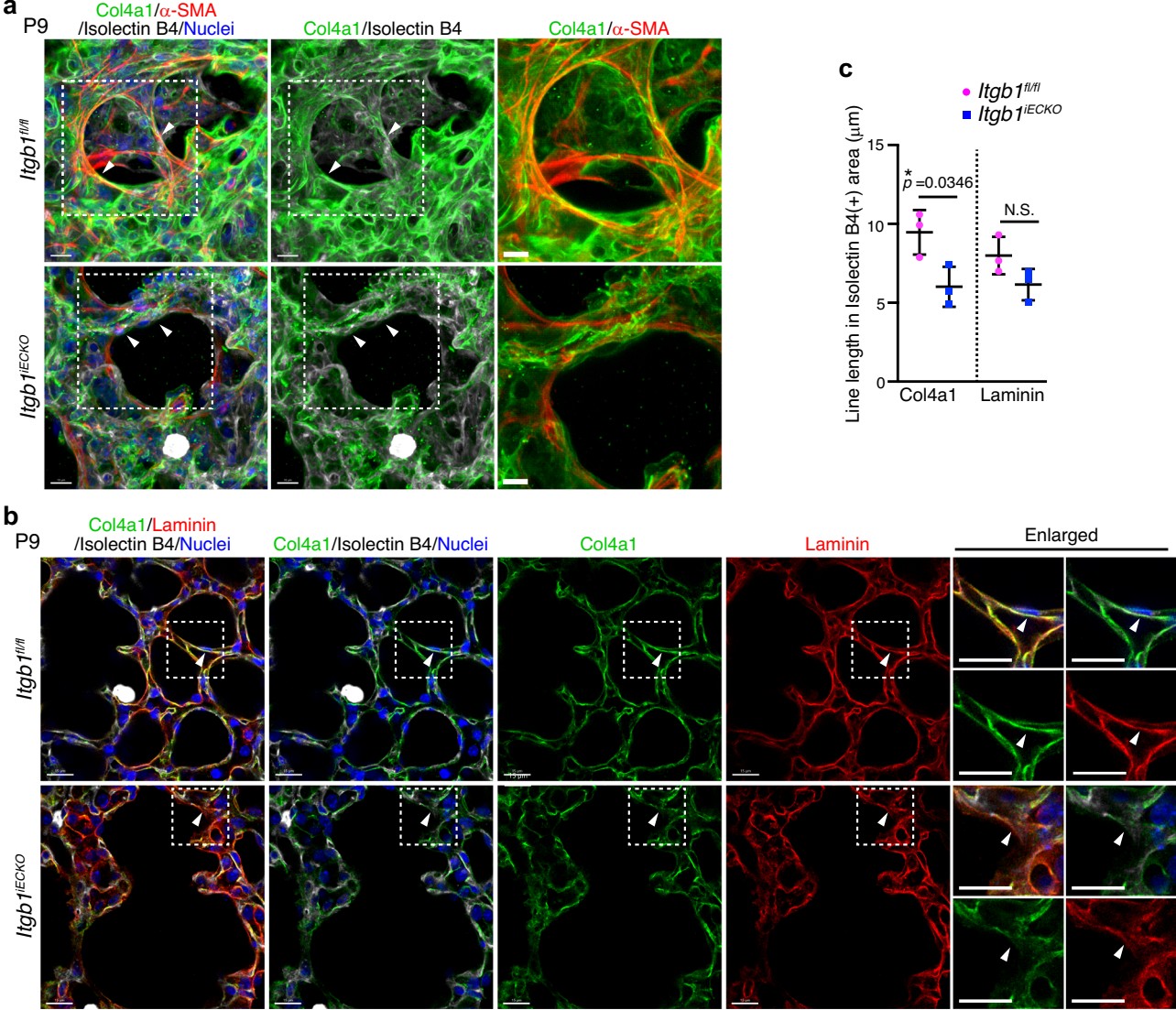

**Fig. 7 | Endothelial integrin β1 regulates formation of BMs required for mechanical signaling leading to myofibroblast contraction. a** Confocal z-projection images for Col4a1 (green), α-SMA (red), isolectin B4 (white), and Nuclei (DAPI, blue) in alveoli from *Itgb1^fl/fl* and *Itgb1^iECKO* mice at P9. Boxed areas are enlarged on the right. Arrowheads, Col4a1-formed BMs surrounding the alveolar myofibroblasts. **b** Single slice fluorescence images for Col4a1 (green), Laminin (red), isolectin B4 (white), and Nuclei (DAPI, blue) in alveoli from *Itgb1^fl/fl* and

*Itgb1^iECKO* mice at P9. Boxed areas are enlarged on the right. Arrowheads, the iso-lectin B4-labeled ECs. **c** Line lengths of Col4a1 (left) and Laminin (right) in isolectin B4(+) areas, as in (**b**). Each dot represents an individual mouse. Data are means ± s.d. (*n* = 3 mice/each). N.S. no significance; *\**P* < 0.05, two-tailed Student's *t*-test (**c**). Scale bars; 10 μm (**a**), 7 μm (enlarged images in **a**), 15 μm (**b**). Source data are provided as a Source data file.

## Cell isolation from lungs

Cell isolation using flow cytometry was performed in accordance with a method described previously[70]. Briefly, the lungs were perfused with ice-cold PBS via the right ventricle of the heart and inflated by injecting 1 ml of 25 U/ml Dispase (Thermo Fisher Scientific) dissolved in PBS through the trachea. The lungs were then excised, followed by removal of bronchi and other tissues. All lobes were transferred into a 6-well plate filled with 5 ml of digestion buffer containing 100 μg/ml Liberase™ (Roche), 0.1 mg/ml DNaseI (Sigma–Aldrich), and 25 mM HEPES, and then extensively minced with scissors. Subsequently, the minced lungs were mixed by pipetting several times using a P1000 pipet equipped with a truncated blue tip, transferred into 50 ml conical tubes, and incubated in a 37 °C water bath for 30 min, during which the lungs were extensively pipetted, every 5 min, using a P1000 pipet tip. After the incubation, 5 ml of FACS buffer (PBS containing 2% fetal calf serum) were added to stop the digestion reaction. Then, the suspended cells were passed through a 100 μm mesh cell strainer and

collected by centrifugation at 330 × *g* for 5 min. The cell pellet was resuspended with 5 ml of red blood cell lysing buffer (Funakoshi) and incubated for 4 min at room temperature (RT). After the incubation, the lysing reaction was stopped by adding 5 ml of FACS buffer. The suspended cells were passed through a 40 μm mesh cell strainer, collected by centrifugation at 330 × *g* for 5 min, and resuspended in 800 μl of FACS buffer.

To achieve sorting of CD31(+) cells and PDGFRα(+) cells, PE-conjugated anti-CD31(BioLegend, 1/200) and APC-conjugated anti-PDGFRα (eBioscience, 1/200) antibodies were used for positive selec-tion, while FITC-conjugated anti-CD45 (BioLegend, 1/200), FITC-conjugated anti-TER119 (BioLegend, 1/200), and FITC-conjugated anti-CD326/EpCAM (BioLegend, 1/200) antibodies were used for negative selection (Supplementary Table 1). For immunostaining, the cells were incubated in FACS buffer containing the above-mentioned antibodies for 30 min on ice with intermittent mixing. After washing with a 10-fold volume of FACS buffer, the cells were resuspended in

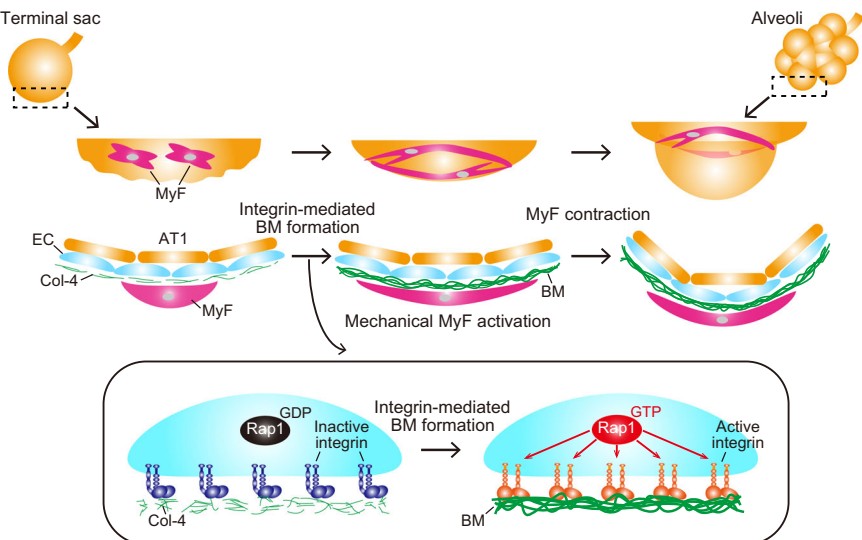

**Fig. 8 | Schematic representation of proposed model accounting for how ECs regulate alveologenesis.** During alveologenesis, alveolar ECs generate BMs by stimulating integrin β1-mediated recruitment of Col-4 in a Rap1-dependent manner. BMs formed by ECs act as a scaffold for alveolar myofibroblasts (MyF) to induce mechanical signal-mediated secondary septa formation, ultimately giving rise to alveoli.

1 ml of FACS buffer containing SYTOX Blue (Thermo Fisher Scientific, 1/500) to label the dead cells. Then, more than $5 \times 10^6$ cells showing FITC(−), APC (−), SYTOX Blue(−), and PE(+) and those showing FITC(−), PE(−), SYTOX Blue(−), and APC(+) were collected as CD31(+) and PDGFRα(+) cell populations, respectively, employing a cell sorter, BD FACSAria™ Fusion (BD biosciences), using a 100 μm nozzle and FACS Diva software. Then, the sorted cells were pelleted by centrifugation at $330 \times g$ for 10 min, and used for the experiments or stored at −80 °C for RNA purification.

### RNA preparation
To extract total RNA from mouse lungs, the lobes excised from P9 mice were homogenized in TRIzol RNA Isolation Reagents (Thermo Fisher Scientific) using a Physcotron homogenizer (MICROTEC CO., LTD.) for 30 s and subjected to RNA isolation steps according to the manufacturer's instructions. The extracted total RNA was further purified using the RNeasy Plus Kit (QIAGEN) and reverse-transcribed using SuperScript™ III Reverse Transcriptase (Thermo Fisher Scientific) together with 10 mM dNTPs, RNase OUT ribonuclease inhibitor (TOYOBO), and random primers (Thermo Fisher Scientific) according to the manufacturer's instructions.

To isolate total RNA from FACS-sorted cells, $0.5–1 \times 10^7$ cells were lysed in 300 μl TRIzol RNA Isolation Reagents by employing extensive pipetting. Then, the cell lysates were mixed with the same volume of 100% EtOH, vortexed vigorously, and subjected to a further purification procedure using a Direct-zol™ RNA MicroPrep kit (Zymo Research) according to the manufacturer's protocol. Purified RNA was reverse-transcribed using a SuperScript VILO cDNA synthesis kit (Thermo Fisher Scientific) according to the manufacturer's instructions.

### qPCR
Quantitative PCR (qPCR) was performed employing a CFX96 Touch Deep Well real-time PCR detection system (Bio-Rad) using KOD SYBR qPCR Mix (TOYOBO) or Thunderbird Next SYBR qPCR Mix (TOYOBO). The primers used for qPCR are listed in Supplementary Table S2.

### 3D immunofluorescence staining of alveoli
3D immunofluorescence staining of the lungs was conducted according to a previously described method[70]. Briefly, P9 mice were

anesthetized with an anesthetic reagent containing 75 μg/ml medetomidine, 400 μg/ml midazolam, and 500 μg/ml butorphanol. After opening the chest cavity, the lungs were refluxed with PBS through the right ventricle to remove blood and subsequently perfused with 1 ml of pre-warmed 6% gelatin in PBS. Then, the lungs were inflated by injecting pre-warmed 1% low melting agarose (LMT) (Nacalai tesque, Inc.) in PBS through the trachea, allowing the filling of alveolar tissues with the gel. After this process, the lungs were immediately chilled with crushed ice for 5 min to solidify the agarose, excised, and fixed in pre-cooled 2% PFA in PBS at 4 °C for 30 min. After confirming that the gel-filled lungs had solidified, the lobes were separated from each other and placed directly on a cutting stage with glue. Then, 250–300 μm-thick sections were cut from the lobes with a VT1200S vibratome (Leica), fixed in 4% PFA in PBS for 1 h at 4 °C, and washed with PBS/NaN₃. Subsequently, the tissue sections were warmed to 55 °C for 30 min in PBS/NaN₃ to melt out the agarose gel. After repeating this step once, the specimens were blocked with blocking solution (5% normal goat serum and 0.25% TritonX-100 in PBS/NaN₃) for 30 min at 4 °C. Then, individual tissue sections were placed in each well of a 24-well plate and immunostained with primary antibodies in 250 μl blocking solution at 4 °C overnight. After washing with PBS/NaN₃ at 4 °C three times, 30 min each time, the tissue sections were further incubated with secondary antibodies in 250 μl blocking solution at 4 °C overnight and washed with 500 μl of PBS/NaN₃ at 4 °C three times, 30 min each time. The primary and secondary antibodies used for 3D immunofluorescence staining of alveoli are listed in Supplementary Table 1. To mount the tissue sections on slides, mending tape with a square hollow for placing specimens was attached to the slides (Matsunami Glass). The tissue samples were placed in the hollows on the slides filled with handmade mounting media containing 2.5% DABCO, 1,4-diazabicyclo[2.2.2]octane, 50 mM Tris−HCl (pH 8.0), and 20% glycerol or aqueous mounting media (Abcam), then covered with coverslips (Matsunami Glass), and sealed with nail polish. The stained alveoli were imaged employing a FLUOVIEW FV3000 confocal microscope (Olympus) equipped with an ×20 water objective lens (XLUMPLFLN) (Olympus) or a ×60 silicone objective lens (UPLSAPOXS2) (Olympus) operated with FLUOVIEW FV31S-SW software (Olympus). The obtained images were processed and analyzed employing Imaris software (Bitplane).

### Flow cytometry-based quantification of myofibroblasts

Alveolar myofibroblasts, their progenitors, and smooth muscle cells in the lungs of postnatal mice were quantified by flow cytometry with modification of a previously described method[37]. The cells were isolated from the lungs of P9 mice following the method described above in "Cell isolation from lungs" section. Then, the cells were collected by centrifugation at $330 \times g$ for 5 min, blocked with FACS buffer containing 2% BSA at 4 °C for 30 min, and stained with APC-conjugated anti-PDGFRα antibody (Thermo Fisher Scientific, 1/100) in FACS buffer for 30 min on ice. After washing with a 10-fold volume of FACS buffer, the cells were fixed and permeabilized with Foxp3 Fixation/Permeabilization Staining Buffer Set (eBioscience™) according to the manufacturer's protocol. Then, the permeabilized cells were incubated with Cy3-conjugated anti-α-SMA antibody (Sigma–Aldrich, 1/200) in FACS buffer for 30 min at RT. After washing with a 10-fold volume of FACS buffer, the cells were analyzed employing a flow cytometer, BD FACSAria™ III or FACSAria™ Fusion (BD Biosciences). The α-SMA(+)/PDGFRα(+) cells, PDGFRα(+) cells, and α-SMA(+) cells were defined as alveolar myofibroblasts, alveolar myofibroblast progenitors, and smooth muscle cells, respectively.

### Western blot analysis

Western blotting was carried out basically according to a protocol described previously[71]. The lungs of P9 mice were harvested from the chest cavity. Then, the lobes were isolated by removing the trachea and bronchi, homogenized in RIPA buffer containing Tris–HCl (pH7.4), 150 mM NaCl, 1% NP-40, 0.1% SDS, and 0.5% DOC for 30 s with a Physcotron homogenizer (MICROTEC CO., LTD.) equipped with a microshaft, NS-4A (MICROTEC CO., LTD.), and centrifuged at 15,000 rpm at 4 °C for 15 min. The supernatants were mixed with $5 \times$ SDS sample buffer containing 50% glycerol, 0.3 M Tris–HCl (pH6.8), 11.5% SDS, 25% β-mercaptoethanol, and 0.5% bromophenol blue (diluted to $1 \times$ concentration), and boiled for 3 min.

For the detection of RAP1A and RAP1B expression in HUVECs, the cells were trypsinized, collected by centrifugation, and lysed in $1 \times$ SDS sample buffer as described above. The samples were supplemented with DNaseI (final concentration, 100 μg/ml, Sigma–Aldrich), boiled for 3 min, and centrifuged at 15,000 rpm for 5 min, and the supernatants were collected. To detect COL4A1 secreted from HUVECs, conditioned media were mixed with $5 \times$ SDS sample buffer to achieve a $1\times$ concentration. To analyze intracellular COL4A1, the HUVECs were collected and lysed in $1 \times$ SDS sample buffer as described above. Then, the samples were mixed with DNaseI, boiled for 3 min, and centrifuged at 15,000 rpm for 5 min, and the supernatants were collected as described above.

Then, the samples were separated by SDS-polyacrylamide gel electrophoresis using TGX Stain-Free Precast Gel (Bio-Rad) or hand casting polyacrylamide gels using 40% Acrylamide/Bis Solution 37.5:1 (Bio-Rad), transferred to Immobilon-P PVDF Transfer Membrane (Merk Millipore) and immunoblotted with anti-YAP (Novus Biologicals, 1/1000), anti-Col4a1 (Merk Millipore, 1/1000), β-actin antibody (BD Bioscience, 1/2000), and Horseradish peroxidase (HRP) -linked 2nd antibodies as listed in Supplementary Table S1. Detection was performed using HRP substrate, Immobilon® Forte (Millipore) or ImmunoStar Reagents (Wako) using Amersham™ ImageQuant™ 800 (IQ800) or ChemiDoc Tough MP Imaging system (Bio-Rad). All uncropped immunoblot scans can be found in the Source Data file.

### Collagen gel contraction assay

PDGFRα(+) cells ($1 \times 10^6$ cells) isolated from the lungs of P7-8 mice were resuspended in 60 μl of growth medium [DMEM (Wako or GIBCO) containing 10% FBS and antibiotics (Wako)]. Collagen gel solution was prepared by mixing 100 μl of 5 mg/ml Rat tail Collagen I (R&D SYSTEMS SYSTEMS), 2.3 μl of 1 N NaOH, and 11.36 μl of 10 × PBS (pH 7.4) and then kept at 4 °C. Then, 68 μl of collagen gel solution,

172 μl of growth medium, and 60 μl of the cell suspension were sequentially added to make 300 μl of cell mixture (final concentration of collagen I, 1 mg/ml). The cell mixture was gently transferred into a 24-well plate, dried for 1 h at 37 °C under 5% $CO_2$, and covered by 500 μl of growth medium. To suppress myosin-mediated cell contraction, 10 μl of M blebbistatin (Sigma–Aldrich) were included in the growth media when indicated. After the culture for 12 h at 37 °C under 5% $CO_2$, the gels containing cells were released from the edge of the wells by using a sterile needle. After further incubation for 72 h, collagen gels containing the cells were fixed with 4% PFA in PBS at RT for 15 min. Images of the gels were acquired with a SZX16 stereomicroscope (Olympus) equipped with a digital camera (Tough, Olympus). To quantify the contractile ability of PDGFRα(+) cells, the area of collagen gels was measured by using Fiji software (NIH).

### Analysis of cellular localization of YAP in PDGFRα(+) cells derived from the lungs

The PDGFRα(+) cells ($1 \times 10^6$ cells) isolated from the lungs of P9 mice were resuspended in 100 μl of growth medium (DMEM containing 10% FBS and antibiotics), plated onto a 35-mm glass-base dish (Iwaki, ASAHI GLASS Company, Ltd.), and cultured for 48 h at 37 °C under 5% $CO_2$. Then, the cells were fixed with 4% PFA in PBS for 15 min at RT, washed with PBS/$NaN_3$, and permeabilized 0.1% TritonX-100/PBS/$NaN_3$ for 5 min at RT. After blocking with 5% normal donkey serum/ PBS/$NaN_3$ for 30 min at RT, the cells were stained with anti-YAP antibody (Novus Biologicals, 1/200) in 5% normal donkey serum/PBS/$NaN_3$ at 4 °C overnight. After washing with PBS/$NaN_3$ three times, the cells were stained with AlexaFluor ™ 488-labeled Goat anti-Rabbit IgG (Thermo Fisher Scientific, 1/500), Cy3-conjugated anti-α-SMA antibody (Sigma–Aldrich, 1/400), and DAPI (Thermo Fisher Scientific, 1/1000) at 4 °C overnight. Subsequently, the cells were washed with PBS/$NaN_3$ three times. Fluorescence images were obtained using a FV1200 confocal inverted microscope equipped with a 60× oil objective lens (UPlanSAPO 60x/1.35na oil Objective, Olympus) and GaAsP photomultiplier tubes operated with FLUOVIEW Ver. 4.2c software (Olympus).

The cellular localization of YAP was quantified using the Fiji software as described in Supplementary Fig. 7b. Total cellular regions and nuclear regions ($ROI^{nuc}$) were determined based on the fluorescence signals derived from α-SMA and DAPI, respectively. The cytoplasmic regions ($ROI^{cell}$) were defined by subtracting $ROI^{nuc}$ from the total cellular regions. Each area size and signal intensity were quantified by the measure function (Area, mean gray value) within overlayed $ROI^{cell}$ and $ROI^{nuc}$. Mean gray value within $ROI^{nuc}$ was used as nuclear intensity. For calculating intensities in cytoplasm, each mean gray value was multiplied by that area size, and then the products were all added as total intensities for $ROI^{cell}$ and $ROI^{nuc}$. Subtracting total intensities and total area sizes of $ROI^{nuc}$ from $ROI^{cell}$, mean gray values for cytoplasm was calculated per fields (Supplementary Fig. 7b).

### Analysis of focal adhesion and focal complex formation in ECs derived from the lungs

The cell suspension from the lungs of P9 mice was prepared as described in "Cell isolation from lungs" section. Subsequently, CD31(+) cells were purified using the MACS system, following the protocol described in the previous report with slight modifications[72]. The lung cells were collected by centrifugation at $330 \times g$ for 5 min and resuspended with 1 ml of MACS buffer [0.1% BSA (Wako)/D-PBS (−) (Nacalai tesque, Inc.)]. Subsequently, the cell suspensions were mixed with 25 μl CD31 Microbeads (Miltenyi Biotec) and incubated for 15 min at RT with gentle mixing on a bioshaker (TAITEC). After adding 5 ml of MACS buffer, the cell suspensions were applied onto MACS® column placed in a magnetic field for positive selection and washed three times with 1 ml of MACS buffer. Then, the retained cells were eluted from the

column by removing it from the separator, collected by centrifugation at $330 \times g$ for 5 min, and resuspended with EC media containing 20% FBS, DMEM (Wako), Penicillin/Streptomycin (Wako), 100 μg/ml heparin, 1 × ECGS (Sigma–Aldrich). Then, the cells were placed on a 24-well plate coated with 0.1% Gelatin (Wako). After culturing for 4 days with a medium change every other day, the cells were replated on a 35-mm glass-base dish coated with Col-1 and cultured for an additional 3 days in EC media.

To analyze formation of FAs and FCs, the cells were fixed with 4% PFA/PBS at RT for 15 min, permeabilized with 0.1% TritonX-100/PBS at RT for 5 min, and washed with PBS. Then, the cells were blocked with PBS containing 2% normal donkey serum at RT for 1 h, and stained with either anti-vinculin (Sigma, 1/400) and anti-ICAM2 (BD biosciences, 1/200) antibodies or anti-activated integrin β1 (BD biosciences, 1/200), anti-vinculin, and anti-PECAM1 antibodies (R&D SYSTEMS SYSTEMS, 1/400) at 4 °C overnight. After washing with PBS five times, vinculin, ICAM2, activated integrin β1, and PECAM1 were visualized by incubating with their corresponding fluorescence-labeled secondary antibodies [AlexaFlor™647 labeled Donkey anti-mouse IgG (Jackson ImmunoResearch, 1/500), AlexaFlor™488 labeled Donkey anti-rat IgG (Jackson ImmunoResearch, 1/500), AlexaFlor™488 labeled Donkey anti-goat IgG (Jackson ImmunoResearch, 1/500)] diluted in blocking buffer at 4 °C overnight. Rhodamine-phalloidin (Thermo Fisher Scientific, 1/500) and DAPI (1/1000) were also included in the reaction buffer to visualize F-actin and nuclei.

Fluorescence images were obtained using a FV1200 confocal inverted microscope as described in the above section. The sizes and numbers of cells and FAs/FCs were calculated by exploiting the surface function equipped in the Imaris software. Surfaces for cell size and FAs/FCs were created based on the positive staining area for ICAM2 and vinculin, respectively, and calculated automatically.

## siRNA-mediated gene knockdown

HUVECs obtained from KURABO were cultured in HuMedia-EG2 (KURABO) as described previously[73]. For siRNA-mediated gene knockdown, $6 \times 10^3$ HUVECs were seeded on 60-mm dishes, cultured for 24 h, and transfected with control siRNA (Dharmacon), two different sets of siRNA mixtures targeting both *RAP1A* and *RAP1B* (Dharmacon) or for *COL4A1* using Lipofectamine™ RNAiMAX Transfection Reagent (Thermo Fischer Scientific) according to the manufacturer's instructions. The target sequences of siRNA used in this study are listed in Supplementary Table 2. siRNA-transfected HUVECs were used for the experiments after 24 h culture.

## Col-4 deposition assay

ECs isolated from the lungs as described above were seeded on a 35-mm glass-base dish coated with Col-1 at the density of $5 \times 10^3$ cells/dish and cultured in EC media for 1-2 days. Then, the cells were fixed with 4% PFA/PBS at RT for 15 min, blocked with PBS containing 5% normal donkey serum at RT for 1 h, and stained with anti-Col4A1 (Millipore, 1/400) and anti-ICAM2 (1/200) antibodies for 24 h at 4 °C. After washing with PBS three times, the cells were stained with Alexa Fluor™488 labeled Donkey antibody against goat IgG (Jackson ImmunoResearch, 1/1000) and Alexa Fluor™647 labeled Donkey antibody against rat IgG (Jackson ImmunoResearch, 1/1000), and DAPI (1/1000) diluted in blocking buffer at 4 °C overnight. In the experiments shown in Supplementary Fig. 4a, the cells were permeabilized with 0.1% TritonX-100/PBS at RT for 5 min to detect cytoplasmic Col4a1.

Similarly, HUVECs transfected with control siRNA or a mixture of *RAP1A* and *RAP1B* siRNAs were plated on a Col-1-coated glass-base dish and cultured in HuMedia-EG2 without or with $MnCl_2$ for 1–2 days. After the fixation, the cells were blocked and stained with anti-Col4A1 (Millipore, 1/400) antibody for 24 h at 4 °C. Subsequently, the cells were stained with Alexa Fluor™488 labeled Donkey antibody against goat IgG (Jackson ImmunoResearch, 1/1000), rhodamine-phalloidin

(1/500), and DAPI (1/1000) diluted in blocking buffer at 4 °C overnight. In the experiments shown in Supplementary Fig. 4e, HUVECs were permeabilized with 0.1% TritonX-100/PBS at RT for 5 min to detect cytoplasmic COL4A1.

Fluorescence images were obtained using a FV1200 confocal inverted microscope equipped with 40× objective lens (UPlanSApo 40x/0.95na Objective, Olympus) or 60× oil objective lens (UPlanSAPO 60x/1.35na oil Objective) and GaAsP photomultiplier tubes operated with FLUOVIEW Ver. 4.2c software. Deposited area of Col4a1 was measured by Fiji software. The images for Col4a1 were converted into 8-bit images, binarized by adjusting threshold at 25, changed scale based on the scale bar, and measured the area by executing Analyze particles. In addition, the images for DAPI and ICAM2 were analyzed to quantify the number of cells and total cellular areas, respectively.

## Matrigel-based in vitro cord formation assay

For siRNA-mediated gene knockdown, $6 \times 10^3$ HUVECs were seeded on 60-mm dishes, and transfected with control siRNA, two different sets of siRNA mixtures targeting both *RAP1A* and *RAP1B* or for *COL4A1*. siRNA-transfected HUVECs were used for the experiments after 24 h culture.

To induce the formation of cord-like structures, $6 \times 10^3$ HUVECs were placed on the Matrigel (BD biosciencess) poured onto a 35-mm glass-base dish, incubated for 45 min to achieve adherence to the gel, and gently covered with 2 ml of growth medium. To inhibit the adhesive activity of integrin β1, 0.5 μg/ml blocking antibody against integrin β1 (mAb13) (Millipore) was added to the growth medium 3 h after cell plating on the Matrigel. The cells were cultured for 36 h to establish cord-like structures, fixed with 4% PFA/PBS at RT for 15 min, and permeabilized with 0.1% TritonX-100/PBS at RT for 5 min. After washing with PBS, the cells were blocked with PBS containing 2% normal donkey serum at RT for 1 h and stained with anti-Col4A1 antibody (Millipore, 1/400) for 3 h at RT. After washing with PBS several times, the cells were stained with Alexa Fluor™488 Donkey antibody against goat IgG (Thermo Fisher Scientific, 1/1000), 4',6'-diamidino-2-phenylindole (DAPI) (Dojindo, 1/1000), and Alexa Fluor™546 Phalloidin (Thermo Fisher Scientific, 1/500). Images were acquired employing a FV3000 or FV1200 confocal upright microscope equipped with a 20× water objective lens (XLUMPLFLN) and GaAsP photomultiplier tubes and operated with FLUOVIEW FV31S-SW software.

To quantify the coverage ratio of endothelial tubes with COL4A1, the positive area stained with anti-COL4A1 or rhodamine-phalloidin was separately delineated using Fiji software. Subsequently, the area positive for COL4A1 was divided by that of rhodamine-phalloidin.

## Quantitative analysis of imaging data

The mean linear intercept (MLI) was measured to quantify alveologenesis in postnatal mice. To calculate MLI, lungs of P14 mice were stained with anti-Col4a1 antibody (Millipore, 1/400), AlexaFluor™ 647 conjugate of Wheat germ agglutinin (WGA) (Thermo Fisher Scientific, 1/500), and Rhodamine or 488-Phalloidin (Thermo Fisher Scientific) as described in "3D immunofluorescence staining of alveoli" section. Proteins reacting with anti-Col4a1 antibody were visualized with AlexaFluor™ 488, 546, 633 or 647-conjugated secondary antibody (Thermo Fisher Scientific or Jackson ImmunoResearch). Fluorescence images were obtained employing a FV3000 confocal microscope equipped with a ×20 water objective lens (XLUMPLFLN) as described in "3D immunofluorescence staining of alveoli" section. Six randomly selected areas per mouse were imaged. MLI was quantified based on the imaging data according to a method described previously[37].

To quantify the fluorescence intensity of pMLC(Ser19) in myofibroblasts, α-SMA-positive areas excluding arteriole regions were extracted from the fluorescence images of α-SMA using the Fiji image processing software package and defined as myofibroblast areas.

Fluorescence intensity of pMLC(Ser19) within myofibroblast areas was measured and expressed as average pixel intensity.

The connectivity of Col4a1 and Laminin was calculated with the Fiji program using the plug-in function of Ridge detection that automatically draws the maximum line based on the fluorescence intensity. The average lengths of continuous signals for Col4a1 or Laminin in alveolar ECs were calculated using the plug-in function designated as Ridge detection function implemented in the Fiji software (NIH). Three slices at the different z positions were selected from 3D images stained with the antibodies against Col4a1, Laminin, and CD31 or Isolectin B4 by use of Imaris software (Bitplane). The signals of Col4a1 and Laminin within the CD31 or Isolectin B4-positive area were extracted using the surface function with which Imaris software is equipped. The images were taken with the snapshot tool, saved in TIFF format, and then quantitatively analyzed with the Fiji software. First, the images were converted in color mode from RGB to 8-bit grayscale. Ridge detection was conducted without binarization. The values of Sigma, Lower Threshold, and Upper Threshold were 1.94, 0.50, and 1.53, respectively. The SLOPE method was not used for this analysis. All lines (270–880 lines) in 3 images per individual mouse were measured and averaged as the Line length. In addition, the connectivity of Col4a1 in the plasma membrane of lung ECs shown in Supplementary Fig. 3f was similarly quantified. The ROI$^{mGFP}$ were applied to the channel images of Col4a1. Then, the connectivity of Col4a1 within the ROI$^{mGFP}$ was calculated using the plug-in function of Ridge detection as described above.

α-SMA-positive cells, nuclear YAP-positive cells, and α-SMA-positive cells exhibiting the nuclear YAP signal were quantified using Imaris software (Bitplane). Numbers of α-SMA-positive cells and nuclear YAP-positive cells were quantified by counting the nuclei within α-SMA-positive areas and within YAP-positive areas, respectively. The nuclei exhibiting both the α-SMA signal and the YAP signal were also counted to quantify the number of α-SMA-positive cells exhibiting the nuclear YAP signal.

The average sizes and numbers of FAs and FCs in CD31(+) cells were quantified as described in "Analysis of focal adhesion and focal complex formation" section.

The quantification of α-SMA bundles was performed with Fiji software as described in Supplementary Fig. 7a. Using an external plug-in, FeatureJ, compute the smallest eigenvalue of Hessian tensor to extract outlines of alveolar rings, using external plug-in, FeatureJ. Gaussian derivatives (the standard deviation σ = 20 pixels) were calculated in the process. The total outlines of alveoli were extracted from the images stained with Cy3-α-SMA employing the constant parameters (Smallest eigenvalue of Hessian tensor, Smoothing scale, 20 pixels). The extracted image further binarized with the adjustment of threshold < −0.16 with a default mode, linearized by using skeletonize and saved as Outlines of α-SMA ring. Aside from that, modifying smoothing scale up to 10 pixels, the bundles of Cy3-α-SMA were extracted using the original image, adjusted threshold < −3.0 in a default mode, linearized by using skeletonize function, and saved the image as Outlines of α-SMA bundles. The length of the both outlines was measured by a manual tracking for each alveolus, i.e., the length of Outlines of α-SMA ring and Outlines of α-SMA bundles. The score was estimated by the division of the length of Outlines of α-SMA bundles by the corresponding Outlines of α-SMA ring in each ring.

The thickness of elastin fibers was quantified as described in Supplementary Fig. 7c. The volume positively stained with AlexaFluor™633-Hydrazide was estimated using Imaris software (Bitplane). Using the projection image from the corresponding 3D images, the area of Hydrazide-positive area was calculated by exploiting Analyze particles function implemented in Fuji software. Then, the volume was divided by the area per field to obtain the total thickness per field. All images used for the quantification were obtained by a FV3000 confocal microscope equipped with a ×20 water objective lens (XLUMPLFLN) with a ×3 magnification.

The ratio of activated to total integrin β1 in the plasma membrane of lung ECs shown in Fig. 4f was quantified as described in the Supplementary Fig. 7d. Initially, fluorescence images of mGFP were converted into an 8-bit scale. Subsequently, mGFP-positive areas were defined through the binarization, setting the lower limit as 30. After establishing a pixel scale, the Analyzed particles function was applied with a lower limit of 1.5, and the ROIs for mGFP (ROI$^{mG}$) were automatically added. The ROI$^{mGFP}$ were applied to both channels for activated integrin β1 and total integrin β1. Intensities within the ROI$^{mGFP}$ were calculated without binarization.

### Statistics and reproducibility

All experiments in this study were independently repeated more than three times with similar results. All confocal z-projection images and single slice images are representative of more than three independent experiments (Figs. 1f and 2a, more than five independent experiments; Figs. 3a, 3d, 4h, and 7a, three independent experiments; Fig. 6c, four independent experiments). All data are presented as means ± SD unless otherwise noted. Statistical analyses were performed employing the unpaired two-tailed Student's *t*-test and one-way ANOVA followed by Turkey's post-hoc test. Survival outcomes in *Rap1$^{iECKO}$* mice were estimated by the Kaplan–Meier method followed by the log-rank test using GraphPad Prism 10 10.0.2 (GraphPad software, Inc). Data were considered statistically significant if the *p-value* was less than 0.05. No significant difference, $p < 0.05$ and $p < 0.01$ are shown as N.S., *, and **, respectively. Detailed statistic information is provided in Supplementary Table 3.

### Reporting summary

Further information on research design is available in the Nature Portfolio Reporting Summary linked to this article.

## Data availability

The authors declare that all data supporting the findings of this study are available within the main text and supplementary materials. Raw data to generate all graphs within the Figures and Supplementary Figs. and uncropped scans of all blots in the Figures and Supplementary Figs. are provided as a Source Data File. Source data are provided with this paper.

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

## Acknowledgements

We thank K. Matsumoto (Nippon Medical School) for technical support in the preparation of paraffin sections, K. Sawamato (Nagoya City University Graduate School of Medical Sciences) and N. Kaneko (Doshisya University) for their aid in the experiments with floxed *Itgb1* mice, Y. Taguchi (BD Biosciences) for supporting the experiments using FACSAria™ Fusion, and A. Sawaguchi (University of Miyazaki) for his advice regarding the experiments requiring electron microscopy. We also thank H. Ichimiya, Y. Matsushita, K. Kato, M. Fukumoto, and M. Sone for excellent technical assistance. This work was supported by Grants-in-Aid for Scientific Research (B) to S.F. (21H02665), Scientific Research (C) to H.W.-T. (23K06325), for Exploratory Research to S.F. (21K19358, 23K18245) from the Japan Society for the Promotion of Science; by FOREST Program from Japan Science and Technology Agency (JST) under Grant Number JPMJFR220T to H.W.-T.; by research grants from Takeda Science Foundation to H.W.-T. and S.F., from the Mochida Memorial Foundation to H.W.-T., from the Cell Science Research Foundation to H.W.-T., from the Naito Foundation to S.F., from Daiichi Sankyo Foundation of Life Science to S.F., from Astellas Foundation for Research on Metabolic Disorders to S.F., from The Uehara Memorial Foundation to S.F., from the TERUMO LIFE SCIENCE FOUNDATION to S.F., from the NOVARTIS Foundation (Japan) for the Promotion of Science to S.F. and H.W.-T.

## Author contributions

H.W.-T. and S.F. conceived and designed the experiments, interpreted the data, and wrote the manuscript. H.W.-T. performed all of the experiments with help from K. Kato, E.O.-N., T.I., T.W., and H.I., and analyzed the data with help from Y.H., K.N., K.T., and T. Miyata supported the experiments for measuring tissue stiffness. K. Kobayashi and T. Murata aided in the experiments for measuring SpO2. Y.K. and R.F. generated the *Cdh5-CreERT2* mouse line and floxed *Itgb1* mice, respectively. N.M. interpreted the data and discussed the research procedures with H.W.-T. and S.F. All authors commented on the manuscript.

## Competing interests

The authors declare no competing interests.

## Additional information

**Haruko Watanabe-Takano** [1]✉, **Katsuhiro Kato** [2], **Eri Oguri-Nakamura**[1], **Tomohiro Ishii** [1], **Koji Kobayashi**[3], **Takahisa Murata**[3], **Koichiro Tsujikawa**[4], **Takaki Miyata** [4], **Yoshiaki Kubota** [5], **Yasuyuki Hanada**[2,6], **Koichi Nishiyama**[6], **Tetsuro Watabe**[7], **Reinhard Fässler** [8], **Hirotaka Ishii** [9], **Naoki Mochizuki** [10] & **Shigetomo Fukuhara** [1]✉

[1]Department of Molecular Pathophysiology, Institute of Advanced Medical Sciences, Nippon Medical School, 1-1-5 Sendagi, Bunkyo-ku, Tokyo 113-8602, Japan. [2]Department of Cardiology, Graduate School of Medicine, Nagoya University, 65 Tsurumai-cho, Showa-ku, Nagoya, Aichi 466-8550, Japan. [3]Department of Animal Radiology, Graduate School of Agricultural and Life Sciences, University of Tokyo, 1-1-1, Yayoi, Bunkyo-ku, Tokyo 113-8657, Japan. [4]Department of Anatomy and Cell Biology, Graduate School of Medicine, Nagoya University, 65 Tsurumai-cho, Showa-ku, Nagoya, Aichi 466-8550, Japan. [5]Department of Anatomy, Keio University School of Medicine, 35 Shinanomachi, Shinjyuku-ku, Tokyo 160-8582, Japan. [6]Laboratory for Vascular and Cellular Dynamics, Department of Medical Sciences, University of Miyazaki, Miyazaki City, Miyazaki 889-1962, Japan. [7]Department of Biochemistry, Graduate, School of Medical and Dental Sciences, Tokyo Medical and Dental University, Tokyo 113-8549, Japan. [8]Department of Molecular Medicine, Max Planck Institute of Biochemistry, Am Klopferspitz 18, 82152 Martinsried, Germany. [9]Department of Anatomy and Neurobiology, Graduate School of Medicine, Nippon Medical School, 1-1-5 Sendagi, Bunkyo-ku, Tokyo 113-8602, Japan. [10]Department of Cell Biology, National Cerebral and Cardiovascular Center Research Institute, 6-1 Kishibe-shimmachi, Suita, Osaka 564-8565, Japan. ✉e-mail: t-haruko@nms.ac.jp; s-fukuhara@nms.ac.jp

