## [Peer Review File · Nature Communications]

Endothelial cells regulate alveolar morphogenesis by constructing basement membranes acting as a scaffold for myofibroblastsREVIEWER COMMENTS

Reviewer #1 (Remarks to the Author):

This is a superficially excellent manuscript which elegantly describes a novel role for endothelial cells in alveologenesis. An endothelial cell-specific Rap conditional knock out mouse line is presented and the lungs are carefully analyzed phenotypically. The authors show that the expected alveolar cell types are specified in the appropriate places. However, that alveologenesis is impaired, likely due to lung-intrinsic changes in mechanical signaling pathways, particularly in the myofibroblasts. The authors conclude that disruption of Rap in the lung endothelium causes non-autonomous changes in the mechanical properties of the developing myofibroblasts. The data presented suggest that the production of Col4a1 by the ECs is disrupted, also the assembly of elastin filaments by the myofibroblasts, and the basement membrane adhesions of the endothelial cells. Mechanistic experiments in HUVECs are presented to link the secretion of Col4a1 by the endothelial cells to integrin activation. These findings are explored genetically *in vivo* by deleting B1-integrin from the developing endothelium, with similar phenotypes to the Rap1 cKO, including a disrupted Col4a1+ basement membrane. The data presented support the novel hypothesis that organization of the developing basement membrane in the alveolar region by the endothelial cells is required for mechanical force transduction and alveologenesis by the myofibroblasts.

The lung analyses and the quantitation performed are excellent. However, I have 2 major concerns:

1. The animal death phenotype (Fig 1a) is not consistent with a lung simplification phenotype being the primary defect in these animals. There are many reports in the literature where blocking alveolar septation does not cause postnatal death (Le Cras et al. 2004; Gao et al. 2022). This suggests that the cKO animals, which have Rap1 deleted in all endothelial cells throughout the body, have another phenotype which causes death. I would like to see growth curves for the cKO animals as there are multiple whole body KO mouse phenotypes where slow growth and defects in the intestine or liver correlate with a lack of postnatal alveologenesis for example, (Chiang et al. 2005). Postnatal growth restriction synergizes with other factors promoting alveolar simplification (Wedgwood et al. 2016). Moreover, there are very clear data showing that adult mouse alveoli are sensitive to levels of nutrition and can undergo simplification and regeneration in response to caloric restriction and refeeding (G. D. Massaro et al. 2002; D. Massaro et al. 2007).

My major concern here is that it is possible that the lung phenotype is secondary to another physiological defect, especially as the alveolar simplification is relatively mild. Ideally, it would be good to see a lung-specific endothelial Rap1 KO, or a lung specific rescue of the phenotype, to confirm that the phenotypes are due to changes in the lung itself.

Another possibility could be that the authors provide detailed growth curves of the animals, intestinal and liver histology, and a thorough discussion of the possibility that there is a non-lung component either exacerbating, or causing, the phenotypes they observe.

2. It is likely from the data presented that all alveolar cells are specified correctly. However, the authors should confirm at ~P5 that the expected cell types are present in the alveoli: epithelium, fibroblast sub-types, endothelial cells (already shown) and nerves (Zhang et al. 2022) are indeed present.

Minor comments

1. The data are in general well-quantitated. However, in some places it is difficult to see clear differences between the control and mutant images. Is it possible to quantitate the elastin levels in 3h and the activated integrin at the membrane in Fig 4 for example?

Chiang, Ming-Ko, Yi-Chun Liao, Yasuko Kuwabara, and Su Hao Lo. 2005. "Inactivation of Tensin3 in

Mice Results in Growth Retardation and Postnatal Lethality." *Developmental Biology* 279 (2): 368–77.

Gao, Feng, Changgong Li, Susan M. Smith, Neil Peinado, Golenaz Kohbodi, Evelyn Tran, Yong-Hwee Eddie Loh, Wei Li, Zea Borok, and Parviz Minoo. 2022. "Decoding the IGF1 Signaling Gene Regulatory Network behind Alveologenesis from a Mouse Model of Bronchopulmonary Dysplasia." *ELife* 11 (October). <https://doi.org/10.7554/eLife.77522>.

Le Cras, T. D., W. D. Hardie, G. H. Deutsch, K. H. Albertine, M. Ikegami, J. A. Whitsett, and T. R. Korfhagen. 2004. "Transient Induction of TGF- α Disrupts Lung Morphogenesis, Causing Pulmonary Disease in Adulthood." *American Journal of Physiology-Lung Cellular and Molecular Physiology* 287 (4): L718–29.

Massaro, Donald, Emma Alexander, Kristin Reiland, Eric P. Hoffman, Gloria Decarlo Massaro, and Linda Biadasz Clerch. 2007. "Rapid Onset of Gene Expression in Lung, Supportive of Formation of Alveolar Septa, Induced by Refeeding Mice after Calorie Restriction." *American Journal of Physiology. Lung Cellular and Molecular Physiology* 292 (5): L1313-26.

Massaro, Gloria Decarlo, Svetlana Radaeva, Linda Biadasz Clerch, and Donald Massaro. 2002. "Lung Alveoli: Endogenous Programmed Destruction and Regeneration." *American Journal of Physiology. Lung Cellular and Molecular Physiology* 283 (2): L305-9.

Wedgwood, Stephen, Cris Warford, Sharleen C. Agvateesiri, Phung Thai, Sara K. Berkelhamer, Marta Perez, Mark A. Underwood, and Robin H. Steinhorn. 2016. "Postnatal Growth Restriction Augments Oxygen-Induced Pulmonary Hypertension in a Neonatal Rat Model of Bronchopulmonary Dysplasia." *Pediatric Research* 80 (6): 894–902.

Zhang, Kuan, Erica Yao, Shao-An Wang, Ethan Chuang, Julia Wong, Liliana Minichiello, Andrew Schroeder, Walter Eckalbar, Paul J. Wolters, and Pao-Tien Chuang. 2022. "A Functional Circuit Formed by the Autonomic Nerves and Myofibroblasts Controls Mammalian Alveolar Formation for Gas Exchange." *Developmental Cell* 57 (13): 1566-1581.e7.

Reviewer #2 (Remarks to the Author):

In their manuscript, Watanabe-Takano et al identify the control of proper basal lamina (BM) assembly as a novel mechanism, alternative to secretion of angiocrine factors, through which vascular endothelial cells (ECs) can regulate the morphogenesis of the parenchyma of an organ, such as the lung. In particular, the authors demonstrate how postnatal deletion of Rap1a/Rap1b or β 1 integrin in ECs prevents proper alveologenesis that depends on the contractile activity of myofibroblasts. The authors propose a model according to which in ECs Rap1a and Rap1b, by regulating β 1 integrins, would allow the incorporation of type IV collagen into the vascular BM, which in turn would act as a substrate on which myofibroblasts would adhere and exert their contractile force necessary for alveolar morphogenesis. The work is potentially interesting; however, there are several critical aspects that need to be clarified, further investigated, or analyzed in a more robust manner. In particular, it is unclear how the mechanistic model of β 1 integrin regulation by Rap1a/b proposed by the authors reconciles with the well-documented and accepted model that the small GTPase Rap1, acting directly or through effectors such as RIAM, stabilizes the conformational activation of talin, which in turn promotes the conformational activation of integrins by interacting with their β subunit and connecting it to the actin cytoskeleton (10.1038/nrm3624 ; 10.1182/blood-2015-12-638700 ; 10.1182/blood.2021013500).

Major issues

1. Page 14, line 20 and page 15, lines 1-6, Fig. 4d-f and Extended data Fig. 4a-b. A main argument of this manuscript, reiterated in the Discussion, is based on the well-known and accepted notion that the small GTPase Rap1 promotes (via talin) the conformational activation of integrins. Yet, in the results the authors state, "However, in Rap1iECKO mice, plasma membrane localization of integrin β 1 and its activated form was nearly absent from alveolar ECs. Instead, integrin β 1 was diffusely localized in the cytosol of alveolar ECs (Fig. 4e, f and 4 Extended data Fig. 4b,c), confirming the decreased activity of integrin β 1 in alveolar ECs of Rap1iECKO mice." The authors therefore claim that their confocal microscopy analysis on tissue shows that the lack of Rap1 in ECs causes not a reduction in β 1 integrin activation, but rather in the localization of total β 1 integrins (in any conformation, active and inactive) on the cell surface. The authors therefore conclude that in pulmonary ECs Rap1 would not promote the activation, but rather the stabilization of the localization of β 1 integrins (both inactive and active) on the cell surface. This conclusion is difficult to reconcile with the literature (10.1038/nrm3624 ; 10.1182/blood-2015-12-638700 ; 10.1182/blood.2021013500). It is my opinion that the low level of resolution of the data obtained from the tissue analyses (shown in Fig. 4 and Extended data Fig. 4) does not allow drawing any solid conclusions about the subcellular localization of β 1 integrins on the plasma membrane or in cytosolic vesicular compartments of ECs. Thus, the conclusions the authors draw are based on non-reliable and solid data.

Second, it is also unclear how the regulation of β 1 integrins in ECs controls the incorporation of Col4a1 protein into the BM. In particular, in the lungs of Rap1iECKO mice, the levels of Col4a1 protein, but not its mRNA, are significantly reduced (Fig. 3). So, the lack of Rap1a/b either inhibits mRNA translation or promotes Col4a1 protein degradation.

The authors need to characterize in greater detail and precision, first, the fundamental question of the relative amounts and subcellular localization of total and active β 1 integrins, and second, that of the possible degradation of Col4a1 protein in ECs.

Quantitative and subcellular analysis of total and active β 1 integrins of vessel ECs should be investigated by staining and quantifying the two receptor conformations simultaneously on the same lung tissue sections of Rap1fl/fl and Rap1iECKO mice. To make statements about possible subcellular localization in the ECs of blood vessels contained in tissue sections, it is necessary to use a high-resolution approach, for example, but not limited to stimulated emission depletion (STED) microscopy.

Both analyses (amount and localization of total/active β 1 integrins; synthesis/degradation of Col4a1 protein) could be investigated in a complementary and more precise manner on ECs isolated from the lungs of Rap1fl/fl and Rap1iECKO mice and cultured in vitro. Careful analysis by high-resolution confocal microscopy on ECs cultured in vitro could give information both on the localization of β 1 integrins in membrane or cytosolic vesicular compartments and on the possible degradation of Col4a1. The latter aspect could be studied, for example, using an inhibitor of lysosomal degradation such as bafilomycin.

2. The authors should test at least in vitro the impact on actin cytoskeleton and YAP translocation in the nucleus of myofibroblasts isolated from lungs and allowed adhering on the BM deposited by ECs isolated from lungs of Rap1fl/fl and Rap1iECKO mice and cultured in vitro (e.g., see <https://www.jove.com/t/55051/a-rapid-scalable-method-for-isolation-functional-study-analysis-cell>). In any case, the authors should at least directly show that myofibroblasts isolated from the lung are able to adhere on type IV collagen and that increasing doses of this extracellular matrix are able to result in proportional translocation of YAP into their nucleus.

3. Page 14, lines 7-13. Fig. 4a-c. As described on page 41, lines 12-15, images documenting the size and number of adhesive sites were acquired with a simple wide field microscope and a Zyla 4.2 PLUS

sCMOS camera (Andor), rather than with a confocal microscope. This type of analysis is not of sufficient quality and does not conform to recognized state-of-the-art standards. These analyses should be repeated on images acquired by confocal microscopy. It is also needed to state what anti-active β 1 integrin antibody was used.

4. Analysis of the impact of Rap1a/b on the ability of ECs to polymerize type IV collagen performed on HUVECs and shown in Fig. 5 and Extended Fig. 5 are not convincing. Matrigel contains type IV collagen and is not an ideal substrate for testing the ability of ECs to polymerize endogenous (autocrine) type IV collagen. The statement on page 15 "Depletion of both RAP1A and RAP1B severely inhibited accumulation of Col4a1 around the cord-like structures" does not agree with the representative images shown in Fig. 5a, which in turn do not agree with the quantification graphs shown in Fig. 5c. In addition, since both a and b isoforms of Rap1 are effectively silenced, a Western blot analysis with a total anti-Rap1 antibody would be useful to show the strong reduction/absence of the protein. As already suggested in point 1, the authors should perform these analyses on ECs isolated from the lungs of Rap1^{fl/fl} and Rap1^{iECKO} mice and cultured in vitro. Also, it is not appropriate for cells to be adhered on Matrigel.

Minor issues

1. Page 9, lines 4-8: "Indeed, alveolar capillaries in Rap1^{iECKO} mice exhibited relatively discontinuous localization of VE-cadherin at cell-cell junctions compared to those in control siblings (Extended data Fig. 1k, arrows)." The level of resolution does not allow this referee to see what the authors claim. It does not seem to me that this kind of data is useful, not least because, as the authors themselves state, "However, vascular hemorrhage was not with these abnormalities in Rap1^{iECKO} mice (Extended data Fig. 1l), suggesting that the barrier function of alveolar capillaries is not severely compromised in Rap1^{iECKO} mice." My opinion is that there is no clear and obvious alteration in the pattern of VE-cadherin and this is consistent with the absence of hemorrhage. Should the authors wish to maintain this claim, they need to document it better, at greater magnification, and quantify it.

2. Page 10, lines 8-11. "However, we noticed that myofibroblasts in Rap1^{iECKO} mice had actin filaments which appeared weak and fragile, while thick and tense actin filaments had developed in alveolar myofibroblasts of control mice (Fig. 2c), pointing to a functional impairment of myofibroblasts in Rap1^{iECKO} mice." Here, as well as in other parts of the manuscript, the authors use terms such as "weak" and "fragile" that instead of describing the fluorescence pattern, give an assessment of its mechanical strength, which was not quantified instead. In this case, the actin filaments of the lung myofibroblasts of Rap1^{iECKO} mice appear simply "thin." The authors need to provide a quantification of this type of alteration.

3. Fig. 2f. It is extremely difficult to appreciate the staining of YAP in a triple image. It is useful to show the corresponding images of YAP alone alongside.

4. Page 10, lines 11-12 and Fig. 3h: "In control mice, the elastic fibers were thick and tense, whereas in Rap1^{iECKO} mice the elastic fibers were thin and less well assembled,....". As mentioned above in point 2, the term "tense" is not appropriate. The authors should also provide a quantification of the intensity of elastin staining.

5. Page 8, line 16: There is a typo in the word "anhydrate". "carbonic anhydrate 4" should be "carbonic anhydrase 4."

6. Page 6, line 5: it is necessary to specify the meaning of the acronym AT1, as well as the acronym of

the specific AT1 cell marker RAGE mentioned in Extended Data Fig. 2.

7. Page 12 lines 17: as mentioned above in point 2, a term such as "fluffy" is not appropriate.

Reviewer #3 (Remarks to the Author):

The study is scientifically rigorous and presents meaningful data regarding how Rap1 plays a role in EC-myofibroblast crosstalk in alveolar morphogenesis. The methodology is well-defined and the use of a reductionist approach gives good insight into the role of the basement membrane ECM components in these pathways. There are only minor concerns I have prior to publication:

1. Fig 4. The quantification shown in Extended Data Fig 4a should be included in the main Figure 4 as it adds quantitative values to the images shown in Fig 4.

2. Limitations of the study are not addressed in the discussion.

Replies to the reviewers' comments

Reviewer #1 (Remarks to the Author):

This is a superficially excellent manuscript which elegantly describes a novel role for endothelial cells in alveologenesis. An endothelial cell-specific Rap conditional knock out mouse line is presented and the lungs are carefully analyzed phenotypically. The authors show that the expected alveolar cell types are specified in the appropriate places. However, that alveologenesis is impaired, likely due to lung-intrinsic changes in mechanical signaling pathways, particularly in the myofibroblasts. The authors conclude that disruption of Rap in the lung endothelium causes non-autonomous changes in the mechanical properties of the developing myofibroblasts. The data presented suggest that the production of Col4a1 by the ECs is disrupted, also the assembly of elastin filaments by the myofibroblasts, and the basement membrane adhesions of the endothelial cells. Mechanistic experiments in HUVECs are presented to link the secretion of Col4a1 by the endothelial cells to integrin activation. These findings are explored genetically in vivo by deleting B1-integrin from the developing endothelium, with similar phenotypes to the Rap1 cKO, including a disrupted Col4a1+ basement membrane. The data presented support the novel hypothesis that organization of the developing basement membrane in the alveolar region by the endothelial cells is required for mechanical force transduction and alveologenesis by the myofibroblasts.

The lung analyses and the quantitation performed are excellent.

However, I have 2 major concerns:

First, we thank reviewer #1 for his/her supportive opinions on our study and insightful comments. We agree that addressing the concerns raised would greatly improve our manuscript. Thus, we have extensively performed the additional experiments and have revised the manuscript accordingly, as described below.

- 1. The animal death phenotype (Fig 1a) is not consistent with a lung simplification phenotype being the primary defect in these animals. There are many reports in the literature where blocking alveolar septation does not cause postnatal death (Le Cras et al. 2004; Gao et al. 2022). This suggests that the cKO animals, which have Rap1 deleted in all endothelial cells throughout the body, have another phenotype*

which causes death. I would like to see growth curves for the cKO animals as there are multiple whole body KO mouse phenotypes where slow growth and defects in the intestine or liver correlate with a lack of postnatal alveologenesis for example, (Chiang et al. 2005). Postnatal growth restriction synergizes with other factors promoting alveolar simplification (Wedgwood et al. 2016). Moreover, there are very clear data showing that adult mouse alveoli are sensitive to levels of nutrition and can undergo simplification and regeneration in response to caloric restriction and refeeding (G. D. Massaro et al. 2002; D. Massaro et al. 2007).

My major concern here is that it is possible that the lung phenotype is secondary to another physiological defect, especially as the alveolar simplification is relatively mild. Ideally, it would be good to see a lung-specific endothelial Rap1 KO, or a lung specific rescue of the phenotype, to confirm that the phenotypes are due to changes in the lung itself.

Another possibility could be that the authors provide detailed growth curves of the animals, intestinal and liver histology, and a thorough discussion of the possibility that there is a non-lung component either exacerbating, or causing, the phenotypes they observe.

As the reviewer pointed out, the defective alveologenesis in *Rap1^{iECKO}* mice could be a secondary outcome of growth retardation and developmental defects in other organs, since *Rap1* is deleted in all endothelial cells (ECs) throughout the body. Indeed, the defective organ development leading to calorie restriction and growth retardation reportedly result in alveolar simplification. Therefore, to exclude the above possibility, we performed additional experiments and conclusively showed that the impaired alveologenesis in *Rap1^{iECKO}* mice is attributable to alterations within the lung ECs themselves. This conclusion is supported by the following results.

- 1) We examined the body weights of control and *Rap1^{iECKO}* mice, and found their growth curves to differ minimally at least until P14, at which timepoint *Rap1^{iECKO}* mice already exhibited hypo-alveolarization (Supplementary Fig. 1e). In addition, we conducted histological analyses of control and *Rap1^{iECKO}* mice, and found that *Rap1^{iECKO}* postnatal mice showed normal development of the liver, stomach, and intestine (Supplementary Fig. 1g). These results indicate that the impaired alveologenesis in *Rap1^{iECKO}* mice is not secondary to growth retardation and/or defective development impacting other organs.
- 2) To further confirm that the alterations in lung ECs result in the impaired alveologenesis in *Rap1^{iECKO}* mice, we performed mosaic deletion of *Rap1a/b* in

ECs by delivering a low dose of tamoxifen through the lactating mother into *Rap1a^{fl/fl};Rap1b^{fl/fl};Cdh5-CreERT2* neonatal mice with the *mTmG* reporter background, which enables ECs lacking *Rap1a/b* with membrane GFP (mGFP) fluorescence to be identified (Supplementary Fig. 3d). We thereby showed that Col4a1 around the mGFP-labeled control ECs was well-assembled to generate continuous basement membranes (BMs), whereas Col4a1 located around the mGFP-labeled *Rap1a/b*-deficient ECs showed less organized assembly, forming discontinuous BMs, suggesting that *Rap1a/b*-deficient ECs fail to induce Col-4 assembly (Supplementary Fig. 3e, f).

Collectively, these results indicate that the hypo-alveolarization observed in *Rap1^{iECKO}* mice is mainly due to functional defects of alveolar ECs, rather than being secondary to their growth retardation or defective development of other organs. These results are now presented as the new Figures and Supplementary Figures. In addition, these observations have been described in the “Results” section with further elaboration in the “Discussion” section.

Additionally, as pointed out by reviewer #1, postnatal death of *Rap1^{iECKO}* mice is unlikely to be due to compromised alveologenesis, since suppression of alveolar septation reportedly does not lead to postnatal death. Thus, further investigation is required to identify the cause of death in *Rap1^{iECKO}* mice. This point is now discussed in the revised “Discussion” section.

- It is likely from the data presented that all alveolar cells are specified correctly. However, the authors should confirm at ~P5 that the expected cell types are present in the alveoli: epithelium, fibroblast sub-types, endothelial cells (already shown) and nerves (Zhang et al. 2022) are indeed present.*

We fully agree with reviewer #1 that the existence (specification) of all cell types involved in alveolarization in *Rap1^{iECKO}* mice requires confirmation, because alveologenesis is a highly coordinated morphogenetic process mediated by multiple cell types. Therefore, we investigated the presence of epithelial cells, myofibroblasts, ECs, and nerves in the terminal sac/alveoli of control and *Rap1^{iECKO}* mice at P4, a timepoint just before the onset of alveolarization. To this end, we conducted 3D immunofluorescence staining of the lungs with Alexa Fluor 633-Hydrazide and anti- α -SMA antibody (for myofibroblasts), anti-RAGE antibody (for epithelial cells), anti-CD31 antibody and Isolectin B4 (for ECs), and anti-neurofilament antibody (for

nerve). We thereby confirmed the presence of all cell types in the terminal sac/alveoli not only in control mice but also in *Rap1^{iECKO}* mice at this timepoint. These results indicate that myofibroblasts, epithelial cells, ECs, and nerves are accurately specified even in the lungs of *Rap1^{iECKO}* mice. Consequently, we conclude that the hypo-alveolarization observed in *Rap1^{iECKO}* mice is not attributable to defective specification of these cell types. These additional data have been included as Supplementary Fig. 2a-c and the “Results” section was revised accordingly.

Minor comments

1. *The data are in general well-quantitated. However, in some places it is difficult to see clear differences between the control and mutant images. Is it possible to quantitate the elastin levels in 3h and the activated integrin at the membrane in Fig 4 for example?*

According to reviewer #1’s suggestion, we quantified imaging data to explicitly indicate the differences between the control and *Rap1^{iECKO}* mice, as described below.

- 1) The thickness of α -SMA filaments in myofibroblasts surrounding the alveolar entry ring was quantified (Fig. 2c, d and Supplementary Fig. 7a). In addition, to clearly illustrate the differences in α -SMA filaments between control and *Rap1^{iECKO}* mice, 3D movies have now been incorporated as Supplementary Movies 1 and 2.
- 2) The thickness of elastin fibers surrounding the alveolar entry ring was quantified (Fig. 3h, i and Supplementary Fig. 7c). In addition, to clearly illustrate the differences in elastin fibers between control and *Rap1^{iECKO}* mice, 3D movies have now been incorporated as Supplementary Movies 3 and 4.
- 3) The plasma membrane localization of activated integrin β 1 in alveolar ECs was quantified (Fig. 4f, g and Supplementary Fig. 7d).
- 4) The duration of continuous signals of Col4a1 surrounding the alveolar ECs was quantified (Supplementary Fig. 3e, f).

These data have been included as the new Figures and Supplementary Figures as indicated above. The corresponding “Results” section was revised accordingly and the detailed quantification protocols are now described in the “Methods” section and outlined in Supplementary Fig. 7.

Reference;

Chiang, Ming-Ko, Yi-Chun Liao, Yasuko Kuwabara, and Su Hao Lo. 2005.
“Inactivation of *Tensin3* in Mice Results in Growth Retardation and Postnatal

Lethality.” *Developmental Biology* 279 (2): 368–77.

Gao, Feng, Changgong Li, Susan M. Smith, Neil Peinado, Golenaz Kohbodi, Evelyn Tran, Yong-Hwee Eddie Loh, Wei Li, Zea Borok, and Parviz Minoo. 2022. “Decoding the IGF1 Signaling Gene Regulatory Network behind Alveologenesis from a Mouse Model of Bronchopulmonary Dysplasia.” *ELife* 11 (October). <https://doi.org/10.7554/eLife.77522>.

Le Cras, T. D., W. D. Hardie, G. H. Deutsch, K. H. Albertine, M. Ikegami, J. A. Whitsett, and T. R. Korfhagen. 2004. “Transient Induction of TGF- α Disrupts Lung Morphogenesis, Causing Pulmonary Disease in Adulthood.” *American Journal of Physiology-Lung Cellular and Molecular Physiology* 287 (4): L718–29.

Massaro, Donald, Emma Alexander, Kristin Reiland, Eric P. Hoffman, Gloria Decarlo Massaro, and Linda Biadasz Clerch. 2007. “Rapid Onset of Gene Expression in Lung, Supportive of Formation of Alveolar Septa, Induced by Refeeding Mice after Calorie Restriction.” *American Journal of Physiology. Lung Cellular and Molecular Physiology* 292 (5): L1313-26.

Massaro, Gloria Decarlo, Svetlana Radaeva, Linda Biadasz Clerch, and Donald Massaro. 2002. “Lung Alveoli: Endogenous Programmed Destruction and Regeneration.” *American Journal of Physiology. Lung Cellular and Molecular Physiology* 283 (2): L305-9.

Wedgwood, Stephen, Cris Warford, Sharleen C. Agvateesiri, Phung Thai, Sara K. Berkelhamer, Marta Perez, Mark A. Underwood, and Robin H. Steinhorn. 2016. “Postnatal Growth Restriction Augments Oxygen-Induced Pulmonary Hypertension in a Neonatal Rat Model of Bronchopulmonary Dysplasia.” *Pediatric Research* 80 (6): 894–902.

Zhang, Kuan, Erica Yao, Shao-An Wang, Ethan Chuang, Julia Wong, Liliana Minichiello, Andrew Schroeder, Walter Eckalbar, Paul J. Wolters, and Pao-Tien Chuang. 2022. “A Functional Circuit Formed by the Autonomic Nerves and Myofibroblasts Controls Mammalian Alveolar Formation for Gas Exchange.” *Developmental Cell* 57 (13): 1566-1581.e7.

We appreciate the reviewer for providing this valuable information.

Reviewer #2 (Remarks to the Author):

In their manuscript, Watanabe-Takano et al identify the control of proper basal lamina (BM) assembly as a novel mechanism, alternative to secretion of angiocrine factors, through which vascular endothelial cells (ECs) can regulate the morphogenesis of the parenchyma of an organ, such as the lung. In particular, the authors demonstrate how postnatal deletion of Rap1a/Rap1b or $\beta 1$ integrin in ECs prevents proper alveologenesi s that depends on the contractile activity of myofibroblasts. The authors propose a model according to which in ECs Rap1a and Rap1b, by regulating $\beta 1$ integrins, would allow the incorporation of type IV collagen into the vascular BM, which in turn would act as a substrate on which myofibroblasts would adhere and exert their contractile force necessary for alveolar morphogenesis. The work is potentially interesting; however, there are several critical aspects that need to be clarified, further investigated, or analyzed in a more robust manner. In particular, it is unclear how the mechanistic model of $\beta 1$ integrin regulation by Rap1a/b proposed by the authors reconciles with the well-documented and accepted model that the small GTPase Rap1, acting directly or through effectors such as RIAM, stabilizes the conformational activation of talin, which in turn promotes the conformational activation of integrins by interacting with their β subunit and connecting it to the actin cytoskeleton (10.1038/nrm3624 ; 10.1182/blood-2015-12-638700 ; 10.1182/blood.2021013500).

We thank reviewer #2 for his/her insightful comments. We agree that addressing the concerns raised would substantially improve our manuscript. Therefore, we have revised the manuscript by addressing all of the concerns raised by reviewer #2, as described in detail below.

Major issues

1. Page 14, line 20 and page 15, lines 1-6, Fig. 4d-f and Extended data Fig. 4a-b. A main argument of this manuscript, reiterated in the Discussion, is based on the well-known and accepted notion that the small GTPase Rap1 promotes (via talin) the conformational activation of integrins. Yet, in the results the authors state, "However, in Rap1iECKO mice, plasma membrane localization of integrin $\beta 1$ and its activated form was nearly absent from alveolar ECs. Instead, integrin $\beta 1$ was diffusely localized in the cytosol of alveolar ECs (Fig. 4e, f and 4 Extended data Fig. 4b,c), confirming the decreased activity of integrin $\beta 1$ in alveolar ECs of Rap1iECKO mice." The authors therefore claim that their confocal microscopy analysis on tissue shows that the lack of

Rap1 in ECs causes not a reduction in $\beta 1$ integrin activation, but rather in the localization of total $\beta 1$ integrins (in any conformation, active and inactive) on the cell surface. The authors therefore conclude that in pulmonary ECs Rap1 would not promote the activation, but rather the stabilization of the localization of $\beta 1$ integrins (both inactive and active) on the cell surface. This conclusion is difficult to reconcile with the literature (10.1038/nrm3624 ; 10.1182/blood-2015-12-638700 ; 10.1182/blood.2021013500). It is my opinion that the low level of resolution of the data obtained from the tissue analyses (shown in Fig. 4 and Extended data Fig. 4) does not allow drawing any solid conclusions about the subcellular localization of $\beta 1$ integrins on the plasma membrane or in cytosolic vesicular compartments of ECs. Thus, the conclusions the authors draw are based on non-reliable and solid data.

Second, it is also unclear how the regulation of $\beta 1$ integrins in ECs controls the incorporation of Col4a1 protein into the BM. In particular, in the lungs of Rap1iECKO mice, the levels of Col4a1 protein, but not its mRNA, are significantly reduced (Fig. 3). So, the lack of Rap1a/b either inhibits mRNA translation or promotes Col4a1 protein degradation.

The authors need to characterize in greater detail and precision, first, the fundamental question of the relative amounts and subcellular localization of total and active $\beta 1$ integrins, and second, that of the possible degradation of Col4a1 protein in ECs.

Quantitative and subcellular analysis of total and active $\beta 1$ integrins of vessel ECs should be investigated by staining and quantifying the two receptor conformations simultaneously on the same lung tissue sections of Rap1fl/fl and Rap1iECKO mice. To make statements about possible subcellular localization in the ECs of blood vessels contained in tissue sections, it is necessary to use a high-resolution approach, for example, but not limited to stimulated emission depletion (STED) microscopy.

Both analyses (amount and localization of total/active $\beta 1$ integrins; synthesis/degradation of Col4a1 protein) could be investigated in a complementary and more precise manner on ECs isolated from the lungs of Rap1fl/fl and Rap1iECKO mice and cultured in vitro. Careful analysis by high-resolution confocal microscopy on ECs cultured in vitro could give information both on the localization of $\beta 1$ integrins in membrane or cytosolic vesicular compartments and on the possible degradation of Col4a1. The latter aspect could be studied, for example, using an inhibitor of lysosomal degradation such as bafilomycin.

We totally agree with reviewer #2 that the following issues needed to be addressed to strengthen our conclusion.

1) Subcellular localization (plasma membrane or cytosolic vesicular compartment) of total and activated form of integrin β 1 in alveolar endothelial cells (ECs).

We fully concur with the reviewer's assessment that the low resolution of the 3D immunofluorescence images of alveoli presented in the previous version of our manuscript did not permit drawing any definitive conclusions regarding the subcellular localization of integrins β 1 in alveolar ECs. Therefore, we carefully analyzed subcellular localizations of both the total and the activated form of integrin β 1 in alveolar ECs and clearly demonstrated the plasma membrane localization of the activated form of integrin β 1 to be significantly diminished in alveolar ECs of *Rap1^{iECKO}* mice, as compared to those of control mice. These observations are described below.

- a. To analyze the plasma membrane localization of the total and the activated form of integrin β 1 in alveolar ECs, we employed a *mTmG* double-fluorescence Cre reporter mouse line, in which the plasma membrane-localized tdTomato is ubiquitously expressed in all cells, while upon Cre-mediated excision of the tdTomato cassette, the plasma membrane-targeted GFP (mG) comes to be expressed in the Cre-targeted cells [*Genesis*, 45(9):593-605 (2007)]. Therefore, we labeled the plasma membranes of alveolar ECs lacking *Rap1a/b* with mG fluorescence by delivering tamoxifen via the lactating mother into the *Rap1a^{fl/fl};Rap1b^{fl/fl};Cdh5-CreERT2* neonatal mice with the *mTmG* reporter background (Supplementary Fig. 3d). Then, we performed 3D immunofluorescence staining of the alveoli with anti-GFP, anti-integrin β 1, and anti-activated integrin β 1 antibodies and acquired the corresponding fluorescence images using a confocal microscope equipped with a \times 60 objective lens (NA 1.35, UPLSAPO \times 60 Oil) (Olympus). Then, we quantified the levels of activated integrin β 1 on the mG-labeled plasma membrane of alveolar ECs as described in the new Supplementary Fig. 7d and the "Methods" section. We thereby demonstrated the plasma membrane localization of the activated form of integrin β 1 to be significantly reduced in *Rap1*-deficient alveolar ECs as compared to control ECs (new Fig. 4f, g). These findings indicate that Rap1 induces the activation of integrin β 1 in alveolar ECs rather than regulating its localization.
- b. According to the reviewer's suggestion, we also investigated the role of Rap1 in integrin-mediated cell adhesions by carefully analyzing the ECs isolated from

control and *Rap1^{iECKO}* mice. For this purpose, ECs were isolated from the lungs of control and *Rap1^{iECKO}* mice, plated onto gelatin-coated dishes, and immunostained with anti-activated integrin β 1, anti-vinculin, and anti-CD31 antibodies. Then, fluorescence images were acquired employing a confocal microscope equipped with a 60 \times objective oil lens (NA 1.35, UPLSAPO \times 60 Oil, Olympus). We thereby demonstrated the number and size of vinculin-labeled focal adhesions (FAs) and focal complexes (FCs) to be reduced in the ECs derived from *Rap1^{iECKO}* mice, as compared to those from control mice (new Fig. 4a, c). Importantly, we also showed the number of activated integrin β 1-positive FAs/FCs to be significantly decreased in lung ECs isolated from *Rap1^{iECKO}* mice, as compared to those from control mice (new Fig. 4d, e). These results indicate that Rap1 induces the activation of integrin β 1 in alveolar ECs.

Collectively, these findings suggest that Rap1 triggers the activation of integrin β 1 to enhance cell adhesions to the extracellular matrix in alveolar ECs. These data have now been included as the new Figures, as indicated above. The corresponding “Results” section has been revised accordingly.

2) How do Rap1 and integrin β 1 control the incorporation of Col4a1 protein into the basement membranes (BMs)? Does the depletion of Rap1 inhibit translation of *Col4a1* mRNA or induce Col4a1 protein degradation?

As the reviewer pointed out, it is necessary for us to elucidate the mechanisms that underlie the Rap1/integrin β 1-induced incorporation of Col4a1 into the BM. To address this question, we carefully examined the ECs isolated from the lungs of control and *Rap1^{iECKO}* mice, as suggested by the reviewer. To achieve this, the isolated ECs were plated onto Collagen I-coated glass bottom dishes, cultured for 72 h, and immunostained with anti-Col4a1 and anti-CD31 antibodies. Then, the fluorescence images of Col4a1 deposited from the ECs were captured employing a confocal microscope equipped with a 60 \times objective oil lens (NA 1.35, UPLSAPO \times 60Oil, Olympus). We observed that Col4a1 was deposited and assembled on the surface of the dish by control lung ECs (new Fig. 5a, b). However, the amount of Col4a1 assembled on the dish was significantly decreased when lung ECs derived from *Rap1^{iECKO}* mice were cultured (new Fig. 5a, b). We also conducted the same experiments using human umbilical vein ECs (HUVECs). Control HUVECs deposited and assembled COL4A1 on the surface of the culture dish (new Fig. 5c, d). However, the depletion of both *RAP1A* and *RAP1B* led to a reduction in COL4A1 assembled on the dish, similarly to the observations made

with lung ECs from *Rap1^{iECKO}* mice (new Fig. 5c, d). Furthermore, we showed that treatment with manganese ions, which increase the ligand-binding affinity of integrins, partially rescued the assembly of COL4A1 by the *RAP1*-depleted HUVECs (Fig. 5c, d). These results suggest that endothelial Rap1 regulates the deposition and/or assembly of Col-4 to form BMs through the activation of integrins. This conclusion is further supported by experiments involving a Matrigel-based in vitro cord formation assay (new Fig. 5g, h), as described below.

We observed the levels of Col4a1 protein, but not its mRNA, to be reduced in the lungs of *Rap1^{iECKO}* mice, as compared to those from control mice (new Fig. 3e, f). As the reviewer pointed out, these results raise the possibility that depletion of Rap1 may induce degradation of Col4a1 or might inhibit the translation of its mRNA. To address this possibility, we analyzed cellular protein levels of COL4A1 in control and *RAP1*-depleted HUVECs as well as those in conditioned media by Western blot analysis. We thereby demonstrated that the cellular protein levels of COL4A1 and the amounts of COL4A1 secreted into the cultured media differed minimally between control and *RAP1*-depleted HUVECs (new Fig. 5e, f), suggesting that the depletion of Rap1 does not lead to Col4a1 degradation and its translation inhibition. These data have been included as the new Figures, as indicated above. The corresponding “Results” section was revised accordingly.

To further confirm this notion, we examined the impact of lysosomal degradation inhibitors, bafilomycin A1 and hydroxychloroquine (HCQ), on the expression levels of COL4A1. Bafilomycin A1 is known to inhibit acidification and protein degradation in lysosomes, and it also disrupts the vesicular transport of

proteins within cells (Dinter and Berger, *Histochem. Cell Biol.* 109: 571-590, 1998; Khine and Sakurai, *Cancers* 15, 2086, 2023). In fact, treatment with bafilomycin A1 and HCQ inhibited the deposition of COL4A1, resulting in the accumulation of cytosolic COL4A1, in both control and *RAP1*-depleted HUVECs (See Figures shown above). Consistently, treatment with bafilomycin A1 and HCQ similarly increased the cellular protein levels of COL4A1 in both control and *RAP1*-depleted HUVECs (See right Figure). These results indicate that depletion of Rap1 does not induce the degradation of Col-4 protein.

Taken together, our data reveal that endothelial Rap1 induces the accumulation of Col-4 around the vessel wall, thereby generating BMs, via the activation of integrins. The remaining question is why the levels of Col4a1 protein are reduced in the lungs of *Rap1^{iECKO}* mice, as compared to those from control mice. It is reasonable to assume that when Col-4 is in an unassembled state, it undergoes digestion by collagenolytic enzymes such as matrix metalloproteinase-2 (MMP2), also known as 72 kDa type IV collagenase. Indeed, a previous study indicate that MMP2 actively induces myofibroblast differentiation during alveologenesi (Watanabe-Takano et al. *PNAS* 111: E2291-2300, 2014), revealing the high activity of MMP2 during alveolarization. Therefore, the reduced quantities of Col-4 protein in the lungs of *Rap1^{iECKO}* mice may result from the digestion of unassembled Col-4 by collagenolytic enzymes. This potential mechanism is now noted in the “Results” section and the need for further investigation in future studies is acknowledged.

2. The authors should test at least *in vitro* the impact on actin cytoskeleton and YAP translocation in the nucleus of myofibroblasts isolated from lungs and allowed adhering on the BM deposited by ECs isolated from lungs of *Rap1^{fl/fl}* and *Rap1^{iECKO}* mice and cultured *in vitro* (e.g., see <https://www.jove.com/t/55051/a-rapid-scalable-method-for-isolation-functional-study-analysis-cell>). In any case, the authors should at least directly show that myofibroblasts isolated from the lung are able to adhere on type IV collagen and that increasing doses of this extracellular matrix are able to result in proportional translocation of YAP into their nucleus.

We fully agree with the reviewer on the importance of confirming that BMs generated by ECs serve as a scaffold for myfibroblasts, thereby stimulating mechanical signaling. Therefore, as suggested by the reviewer, we isolated alveolar myfibroblasts from control and *Rap1^{iECKO}* mice, plated them on glass bottom dishes without coating as well as on dishes coated with Col-4, and analyzed subcellular localizations of YAP and α -SMA filaments by immunostaining with anti-YAP and anti- α -SMA antibodies, respectively. We thereby demonstrated that alveolar myfibroblasts isolated from both control and *Rap1^{iECKO}* mice exhibited nuclear localization of YAP and formation of α -SMA filaments, regardless of the presence or absence of Col-4 coating (Supplementary Fig. 2d, e and See Figures shown below). These results, at a minimum, indicate that alveolar myfibroblasts in *Rap1^{iECKO}* mice maintain the ability to activate mechanical signaling. However, this experiment does not allow us to draw a conclusion regarding the significance of the BM formed by ECs in the activation of mechanical signaling in alveolar myfibroblasts, because myfibroblasts exhibited YAP nuclear localization even when plated on the uncoated dish. Since substrate stiffness induces YAP activation (nuclear localization), it is reasonable to assume that the stiffness of glass bottom dishes may be sufficient to activate mechanical signaling in myfibroblasts. Therefore, it remains difficult to elucidate the role of the BM formed by ECs in activating mechanical signaling in alveolar myfibroblasts.

In addition, our results do not as yet clarify how myfibroblasts sense the stiffness of EC-generated BMs. Cells sense substrate stiffness via integrin-mediated cell adhesion complexes, which link the extracellular matrix to the actin cytoskeleton (Martino et al. Front. Physiol. 9: 824, 2018). Therefore, myfibroblasts might sense the BM stiffness by adhering to BM components such as Col-4 via integrins. In addition, a prior study showed that BMs promote rapid and robust fibronectin assembly using sliding FAs driven by a contractile winch, thereby leading to accumulation of

fibronectin at BMs (Lu et al. Dev. Cell 52: 631-646, 2020). It has also been reported that alveolar myfibroblasts adhere to fibronectin via $\alpha5\beta1$ integrins, which is required for alveologensis (Hrycaj et al. PNAS 115: E10605-E10614, 2018). Thus, myfibroblasts might sense the stiffness of BMs through integrin-mediated adhesion to fibronectin assembled on the EC-generated BMs. These hypotheses are now elaborated upon in the “Discussion” section, and we plan to investigate the mechanism by which myfibroblasts sense the stiffness of EC-generated BMs in future studies.

3. Page 14, lines 7-13. Fig. 4a-c. As described on page 41, lines 12-15, images documenting the size and number of adhesive sites were acquired with a simple wide field microscope and a Zyla 4.2 PLUS sCMOS camera (Andor), rather than with a confocal microscope. This type of analysis is not of sufficient quality and does not conform to recognized state-of-the-art standards. These analyses should be repeated on images acquired by confocal microscopy. It is also needed to state what anti-active $\beta1$ integrin antibody was used.

According to the reviewer’s suggestion, we investigated the adhesive activity of integrins in ECs of control and *Rap1^{iECKO}* mouse lungs by acquiring immunofluorescence images using a confocal microscope. For this purpose, the ECs isolated from the lungs of control and *Rap1^{iECKO}* mice were immunostained with anti-vinculin and anti-activated integrin $\beta1$ antibodies. Then, the immunofluorescence images of vinculin and activated integrin $\beta1$ were acquired employing a confocal microscope equipped with a 60 \times oil objective lens (NA 1.35, UPLSAPO \times 60 Oil, Olympus). We thereby clearly demonstrated the number and size of vinculin-labeled focal adhesions (FAs) and focal complexes (FCs) to be reduced in the ECs derived from *Rap1^{iECKO}* mice, as compared to control ECs (new Fig. 4a, c). In addition, the number of activated integrin $\beta1$ -positive FAs/FCs was significantly decreased in the lung ECs isolated from *Rap1^{iECKO}* mice compared to those from control mice (new Fig. 4d, e). These results indicate that Rap1 is required for activation of integrins in lung ECs. These data are now included as the new Figures, as indicated above. The corresponding “Results” section was revised accordingly.

4. Analysis of the impact of *Rap1a/b* on the ability of ECs to polymerize type IV collagen performed on HUVECs and shown in Fig. 5 and Extended Fig. 5 are not convincing. Matrigel contains type IV collagen and is not an ideal substrate for testing the ability of ECs to polymerize endogenous (autocrine) type IV collagen. The statement on page 15

"Depletion of both RAPIA and RAP1B severely inhibited accumulation of Col4a1 around the cord-like structures" does not agree with the representative images shown in Fig. 5a, which in turn do not agree with the quantification graphs shown in Fig. 5c. In addition, since both a and b isoforms of Rap1 are effectively silenced, a Western blot analysis with a total anti-Rap1 antibody would be useful to show the strong reduction/absence of the protein. As already suggested in point 1, the authors should perform these analyses on ECs isolated from the lungs of Rap1^{fl/fl} and Rap1^{iECKO} mice and cultured in vitro. Also, it is not appropriate for cells to be adhered on Matrigel.

According to reviewer #2's suggestion, we performed the additional experiments to investigate the impact of Rap1 on the ability of ECs to polymerize Col-4 as described above. In these experiments, the ECs isolated from the lungs of control and Rap1^{iECKO} mice were plated onto Collagen I-coated glass bottom dishes, cultured for 72 h, and immunostained with anti-Col4a1 antibody to analyze Col-4 polymerization. We observed that Col4a1 was deposited and assembled on the surface of the dish by control lung ECs (new Fig. 5a, b). However, the amount of Col4a1 assembled on the dish was significantly decreased when lung ECs derived from Rap1^{iECKO} mice were cultured (new Fig. 5a, b). We also conducted the same experiments using control and RAPIA/RAP1B-depleted HUVECs. First, we confirmed efficient knockdown of RAPIA/RAP1B in HUVECs transfected with the corresponding siRNAs by conducting qPCR analysis and Western blot analysis (new Supplementary Fig. 4b, c). Then, we found that control HUVECs deposited and assembled COL4A1 on the surface of the culture dish, while the depletion of both RAPIA and RAP1B led to a reduction in COL4A1 assembled on the dish, similar to the observations made with lung ECs from Rap1^{iECKO} mice (new Fig. 5c, d). Furthermore, we showed treatment with manganese ions, which increase the ligand-binding affinity of integrins, to partially rescue the assembly of COL4A1 by the RAPI-depleted HUVECs (new Fig. 5c, d). Considering these observations, we concluded that Rap1 actively induces the assembly of Col-4 via the activation of integrins in lung ECs. These data have now been included as the new Figures, as indicated above. The corresponding "Results" section was revised accordingly.

Regarding the experiments to assess the ability of ECs to polymerize COL-4 using Matrigel, we agree with the reviewer that the representative images shown in the previous version of our manuscript were not convincing. Therefore, we conducted the additional experiments and have replaced the previous images with new ones that more

clearly highlight the differences between the control and knockdown groups (new Fig. 5g, h). These images, especially the cross-sectional single-plane images in Fig. 5g, clearly show the accumulation COL4A1 around the vessel wall to be significantly reduced in *RAP1*-depleted HUVECs, as compared to control HUVECs. Furthermore, this reduced accumulation was partially rescued by treatment with manganese ions.

As the reviewer pointed out, Matrigel is a mixture of extracellular matrix containing Col-4. However, we showed that the coverage of cord-like structures by COL4A1 was completely inhibited by siRNA-mediated knockdown of COL4A1 in HUVECs, suggesting that HUVECs produce and assemble COL4A1 around the vessel wall to form BMs. Therefore, the reduced coverage of cord-like structures by COL4A1 in *RAP1*-depleted HUVECs is a consequence of the impaired assembly of COL4A1 secreted from HUVECs. These findings suggest that Rap1 induces the accumulation of EC-secreted Col-4 around vessel walls to form BM. These data have now been included as the new Figures, as indicated above. The corresponding “Results” section was revised accordingly.

Minor issues

1. Page 9, lines 4-8: *"Indeed, alveolar capillaries in Rap1^{iECKO} mice exhibited relatively discontinuous localization of VE-cadherin at cell-cell junctions compared to those in control siblings (Extended data Fig. 1k, arrows)." The level of resolution does not allow this referee to see what the authors claim. It does not seem to me that this kind of data is useful, not least because, as the authors themselves state, "However, vascular hemorrhage was not with these abnormalities in Rap1^{iECKO} mice (Extended data Fig. 1l), suggesting that the barrier function of alveolar capillaries is not severely compromised in Rap1^{iECKO} mice." My opinion is that there is no clear and obvious alteration in the pattern of VE-cadherin and this is consistent with the absence of hemorrhage. Should the authors wish to maintain this claim, they need to document it better, at greater magnification, and quantify it.*

We agree with the reviewer that the previous Extended data Fig. 1k did not convincingly support our statement: “Indeed, alveolar capillaries in *Rap1^{iECKO}* mice exhibited relatively discontinuous localization of VE-cadherin at cell-cell junctions compared to those in control siblings”. As the reviewer pointed out, we did not observe vascular hemorrhage in the alveolar capillaries of *Rap1^{iECKO}* mice (new Supplementary Fig. 1n), suggesting preserved barrier function in these capillaries. Since these findings already indicate that the defective alveologenesis in *Rap1^{iECKO}* mice is not caused by

vascular hemorrhage in alveolar capillaries, we removed the previous Extended data Fig. 1k depicting the localization of VE-cadherin at cell-cell junctions in alveolar capillaries.

2. Page 10, lines 8-11. *"However, we noticed that myofibroblasts in Rap1iECKO mice had actin filaments which appeared weak and fragile, while thick and tense actin filaments had developed in alveolar myofibroblasts of control mice (Fig. 2c), pointing to a functional impairment of myofibroblasts in Rap1iECKO mice." Here, as well as in other parts of the manuscript, the authors use terms such as "weak" and "fragile" that instead of describing the fluorescence pattern, give an assessment of its mechanical strength, which was not quantified instead. In this case, the actin filaments of the lung myofibroblasts of Rap1iECKO mice appear simply "thin." The authors need to provide a quantification of this type of alteration.*

As the reviewer pointed out, the use of ambiguous terms like “weak” and “fragile” to characterize the appearance of α -SMA filaments in myofibroblasts *Rap1^{iECKO}* mice (new Fig. 2c) is not appropriate. Instead, it is necessary to provide quantification data describing the fluorescence pattern. Therefore, we quantified the thickness of α -SMA filaments in myofibroblasts of control and *Rap1^{iECKO}* mice as described in the “Methods” section and Supplementary Fig. 7a and showed that the filaments of α -SMA in myofibroblasts of *Rap1^{iECKO}* mice were significantly thinner than those of controls (new Fig. 2d). Accordingly, the preceding sentence was revised as follows: “However, we observed that the α -SMA filaments in myofibroblasts were markedly thinner in *Rap1^{iECKO}* mice than in controls (Fig. 2c, d and Supplementary Movie1, 2), pointing to a functional impairment of myofibroblasts in *Rap1^{iECKO}* mice”. Additionally, we have carefully revised similar sentences throughout the manuscript.

3. Page 10, lines 11-12 and Fig. 3h: *"In control mice, the elastic fibers were thick and tense, whereas in Rap1iECKO mice the elastic fibers were thin and less well assembled,....". As mentioned above in point 2, the term "tense" is not appropriate. The authors should also provide a quantification of the intensity of elastin staining.*

We agree with the reviewer, as stated in point 2. Hence, we quantified the thickness of elastin fibers surrounding the alveolar entry ring in control and *Rap1^{iECKO}* mice as described in the “Methods” section and Supplementary Fig. 7c, and found that the elastin fibers in *Rap1^{iECKO}* mice were significantly thinner than those of controls (Fig.

3h, i). Accordingly, the preceding sentence was revised as follows: “However, quantitative analysis revealed that the elastic fibers surrounding the alveolar entry ring were thinner in *Rap1^{iECKO}* mice than in control mice”. Additionally, we have carefully revised similar sentences throughout the manuscript.

4. *Page 8, line 16: There is a typo in the word "anhydrate". "carbonic anhydrate 4" should be "carbonic anhydrase 4."*

Thank you for pointing out this typographical error, which has now been corrected.

5. *Page 6, line 5: it is necessary to specify the meaning of the acronym AT1, as well as the acronym of the specific AT1 cell marker RAGE mentioned in Extended Data Fig. 2*

We now spell out these acronyms at the first use in our manuscript.

6. *Fig. 2f. It is extremely difficult to appreciate the staining of YAP in a triple image. It is useful to show the corresponding images of YAP alone alongside.*

Following the reviewer’s suggestion, we have included individual images of YAP alone alongside, within the new Fig. 2g.

7. *Page 12 lines 17: as mentioned above in point 2, a term such as "fluffy" is not appropriate.*

Following the reviewer’s suggestion, we have revised the manuscript by avoiding ambiguous terms such as “fluffy”, as stated in response points 2 and 3.

Reviewer #3 (Remarks to the Author):

The study is scientifically rigorous and presents meaningful data regarding how Rap1 plays a role in EC-myofibroblast crosstalk in alveolar morphogenesis. The methodology is well-defined and the use of a reductionist approach gives good insight into the role of the basement membrane ECM components in these pathways. There are only minor concerns I have prior to publication:

First, we thank reviewer #3 for his/her positive opinions on our study and the valuable comments provided. According to the reviewer's comments, we have revised our manuscript as described below. We are confident that this revision has improved our manuscript.

1. Fig 4. The quantification shown in Extended Data Fig 4a should be included in the main Figure 4 as it adds quantitative values to the images shown in Fig 4.

According to reviewer #3's suggestion, the previous Extended Data Fig 4a showing the number and size of focal adhesions and focal complexes has now been included as the new main Fig. 4c.

2. Limitations of the study are not addressed in the discussion.

According to reviewer #3's comment, "Limitations of the study" is now in the "Discussion" section.

REVIEWERS' COMMENTS

Reviewer #1 (Remarks to the Author):

The authors have done a good job of addressing all of my concerns. I am impressed by many of the experiments that they have done to address reviewer 2's concerns also.

Reviewer #2 (Remarks to the Author):

The authors rigorously carried out a number of experiments that clearly addressed the questions and concerns that arose in the review process.